# Selenoprotein W ensures physiological bone remodeling by preventing hyperactivity of osteoclasts

Hyunsoo Kim[1,2,12], Kyunghee Lee[1,12], Jin Man Kim[1], Mi Yeong Kim[1], Jae-Ryong Kim[3], Han-Woong Lee[4], Youn Wook Chung[5], Hong-In Shin[6], Taesoo Kim[7], Eui-Soon Park[8], Jaerang Rho[8], Seoung Hoon Lee[9], Nacksung Kim[10], Soo Young Lee[11], Yongwon Choi[2] & Daewon Jeong[1✉]

Selenoproteins containing selenium in the form of selenocysteine are critical for bone remodeling. However, their underlying mechanism of action is not fully understood. Herein, we report the identification of selenoprotein W (SELENOW) through large-scale mRNA profiling of receptor activator of nuclear factor (NF)-κB ligand (RANKL)-induced osteoclast differentiation, as a protein that is downregulated via RANKL/RANK/tumour necrosis factor receptor-associated factor 6/p38 signaling. RNA-sequencing analysis revealed that SELENOW regulates osteoclastogenic genes. *SELENOW* overexpression enhances osteoclastogenesis in vitro via nuclear translocation of NF-κB and nuclear factor of activated T-cells cytoplasmic 1 mediated by 14-3-3γ, whereas its deficiency suppresses osteoclast formation. *SELENOW*-deficient and *SELENOW*-overexpressing mice exhibit high bone mass phenotype and osteoporosis, respectively. Ectopic *SELENOW* expression stimulates cell-cell fusion critical for osteoclast maturation as well as bone resorption. Thus, RANKL-dependent repression of *SELENOW* regulates osteoclast differentiation and blocks osteoporosis caused by overactive osteoclasts. These findings demonstrate a biological link between selenium and bone metabolism.

[1] Department of Microbiology, Laboratory of Bone Metabolism and Control, Yeungnam University College of Medicine, Daegu, Korea. [2] Departments of Pathology and Laboratory Medicine, University of Pennsylvania School of Medicine, Philadelphia, PA, USA. [3] Department of Biochemistry and Molecular Biology, Smart-aging Convergence Research Center, Yeungnam University College of Medicine, Daegu, Korea. [4] Department of Biochemistry, College of Life Science and Biotechnology, Yonsei University, Seoul, Korea. [5] Severance Biomedical Science Institute, Yonsei University College of Medicine, Seoul, Korea. [6] IHBR, Department of Oral Pathology, School of Dentistry, Kyungpook National University, Daegu, Korea. [7] Herbal Medicine Research Division, Korea Institute of Oriental Medicine, Daejeon, Korea. [8] Department of Microbiology and BK21 Bio Brain Center, Chungnam National University, Daejeon, Korea. [9] Department of Oral Microbiology and Immunology, Wonkwang University School of Dentistry, Iksan, Korea. [10] National Research Laboratory for Regulation of Bone Metabolism and Disease, Chonnam National University Medical School, Gwangju, Korea. [11] Division of Life and Pharmaceutical Sciences, Department of Life Science, Center for Cell Signaling & Drug Discovery Research, College of Natural Sciences, Ewha Womans University, Seoul, Korea. [12] These authors contributed equally: Hyunsoo Kim, Kyunghee Lee. ✉email: dwjeong@ynu.ac.kr

Cellular functions are regulated by positive (feed-forward) or negative feedback[1–3]. The balance between these processes determines cell fate and its perturbation can lead to cellular malfunction and a switch from a normal to a pathological state. In bone physiology, dysregulation of the bone-forming and -degrading activities of osteoblasts and osteoclasts, respectively, results in abnormal bone remodelling. For instance, osteoporosis occurs due to osteoclast overactivity[4]. Proper differentiation of mononuclear hematopoietic progenitors of the myeloid lineage into multi-nucleated osteoclasts is achieved through upregulation of receptor activator of nuclear factor (NF)-κB ligand (RANKL)-induced positive factors [e.g. nuclear factor of activated T-cells, cytoplasmic (NFATc)1; osteoclast-associated, immunoglobulin-like receptor (OSCAR); ATPase H$^+$-transporting V0 subunit D2 (Atp6v0d2), dendrocyte-expressed seven-transmembrane protein (DC-STAMP); microphthalmia-associated transcription factor; and c-Fos[5–10] and downregulation of negative factors [e.g. inhibitor of DNA binding (Id)2, V-maf musculoaponeurotic fibrosarcoma oncogene homologue (Maf)B, interferon regulatory factor (IRF)8, B cell lymphoma (Bcl)6, and LIM homeobox 2][11–15]. Many signalling networks are known to govern osteoclast fate determination. However, the observation that a RANKL-induced downregulated factor stimulates osteoclastogenesis has yet to be fully explained.

Selenium is an essential trace element that serves as a bone-building mineral[16] and is required for the biosynthesis of selenoproteins, which contain selenocysteine (SeCys) encoded by a UGA codon that is normally recognised as a stop codon[17]. There are 25 and 24 known selenoproteins in humans and rodents, respectively[18,19]. Most of these—particularly glutathione peroxidases and thioredoxin reductases—have an important role in maintaining cellular antioxidant homoeostasis[20]. Additionally, some selenoproteins of unknown function (SELENOH, SELENOM, SELENOF, SELENOT, SELENOV and SELENOW) harbour a thioredoxin-like domain with a CysXXSeCys redox motif (where X is any amino acid), implying that they have an antioxidant function[21]. Nutritional selenium deficiency or genetic abnormalities in selenoproteins are associated with endocrine defects, including cretinism, thyroid hormone default, osteoarthritis (termed Kashin–Beck disease), and growth retardation caused by delayed bone formation[22–24]. Selenium status is positively correlated with bone mineral density in healthy ageing males[25] and post-menopausal women[26], and mutations in selenocysteine insertion sequence (SECIS)-binding protein 2 lead to defective selenoprotein biosynthesis, which manifests as delayed skeletal development and linear growth[27]. Mice deficient in SeCys tRNA, which is required for the biosynthesis of all selenoproteins, and SeCys-rich selenoprotein P, which is responsible for selenium transport and storage, exhibit abnormal skeletal development[28,29]. Although some studies have suggested a connection between selenium, selenoproteins, and bone metabolism, there are no known selenoproteins to date that participate exclusively in bone remodelling.

In the present study, we identified, through a large-scale mRNA profiling analysis, selenoprotein W (SELENOW)[30], a protein of unknown function containing a SeCys encoded by UGA at codon 13 whose expression is negatively regulated by RANKL. SELENOW was originally reported as being associated with the white coloration in selenium-deficient regions of calcified cardiomyopathy[30]. Here, we show that SELENOW acts as an osteoclastogenic stimulator and engages in negative feedback to suppress osteoclast differentiation and the pathological shift towards bone disorders.

## Results

**SELENOW stimulates osteoclastogenesis.** To identify novel genes that are up- or downregulated during RANKL-induced osteoclastogenesis, we carried out mRNA expression profiling using GeneChip arrays. We identified several genes known to be upregulated [e.g. *Calcr* (encoding calcitonin receptor), *Ctsk* (encoding cathepsin K), *OSCAR*, *Itgb3* (encoding integrin β3), *c-Fos*, and *NFATc1*][31,32] or downregulated (e.g. *Id2* and *IRF8*)[11,13] by RANKL that positively and negatively regulate osteoclastogenesis, respectively (Supplementary Fig. 1a, b). Among the downregulated genes, we focused on SELENOW, a ~10-kDa protein that is ubiquitously expressed, with especially high levels detected in the brain, liver, skeletal muscle, and long bone (Supplementary Fig. 1c, d).

*SELENOW* expression was gradually increased during the differentiation of macrophage colony-stimulating factor (M-CSF)-induced bone marrow-derived macrophages, which are osteoclast precursors (Supplementary Fig. 2a). In contrast, *SELENOW* levels were decreased from the initiation of RANKL-induced osteoclastogenesis in the presence of M-CSF, despite the increase in expression of several osteoclastic-specific markers, including *Acp5* (encoding tartrate-resistant acid phosphatase; TRAP), *NFATc1*, *OSCAR*, *c-Fos*, *cathepsin K*, and *DC-STAMP* in differentiating cells (Fig. 1a and Supplementary Fig. 2a). *SELENOW* repression was confirmed during the differentiation of RAW264.7 cells into osteoclasts in response to RANKL stimulation (Supplementary Fig. 2b). These results indicate that SELENOW downregulation during osteoclastogenesis is dependent on RANKL signalling.

Many studies have reported that binding of RANKL to its cognate receptor RANK on the osteoclast precursor membrane leads to the recruitment of tumour necrosis factor receptor-associated factor (TRAF)6, which mediates downstream signals that promote osteoclastogenesis, involving mitogen-activated protein kinases (MAPKs) and transcription factors such as NF-κB, activator protein (AP)-1, and NFATc1[33,34]. To clarify the mechanism underlying RANKL-dependent inhibition of SELENOW, we evaluated the regulation of SELENOW expression by factors downstream of RANKL/RANK. Treatment with interferon-γ, which induces rapid proteasomal degradation of TRAF6[35] and suppresses osteoclastogenesis by blocking the RANKL-RANK signalling, failed to induce RANKL-mediated *SELENOW* downregulation (Fig. 1b). This was confirmed by examining the differentiation of TRAF6-deficient osteoclasts (Fig. 1c and Supplementary Fig. 2c), which indicated that *SELENOW* expression is inhibited via the RANKL/RANK/TRAF6 axis. Additionally, RANKL-induced *SELENOW* suppression was enhanced by the MAPK kinase inhibitor PD98059, an effect that was reversed by the p38 inhibitor SB203580 (Fig. 1d). On the other hand, RANKL-induced *SELENOW* downregulation was unaffected by inhibitors of c-Jun N-terminal kinase (JNK; SP600125), NF-κB (SN50), and NFATc1 [cyclosporin A (CsA)]. RANKL-induced p38 activation was abolished in TRAF6-deficient cells, although extracellular signal-regulated kinase (ERK) was activated by RANKL irrespective of TRAF6 expression (Supplementary Fig. 2d, e). These results indicate that *SELENOW* expression is induced by RANKL/RANK/ERK and blocked by RANKL/RANK/TRAF6/p38 signalling.

To investigate the role of SELENOW in osteoclastogenesis, we silenced and overexpressed it using a lentivirus carrying a short hairpin (sh) RNA and a retroviral gene induction system, respectively. *SELENOW* knockdown decreased whereas its overexpression increased osteoclast differentiation (Fig. 1e, f). However, *SELENOW* expression was unaltered during osteoblast differentiation and modulating *SELENOW* levels had no effect on this process or on the expression of osteoblast differentiation markers such as *Alp* (encoding alkaline phosphatase) and *Spp1* (encoding osteopontin)[36] (Supplementary Fig. 3). These results indicate that SELENOW acts as a positive regulator of osteoclast

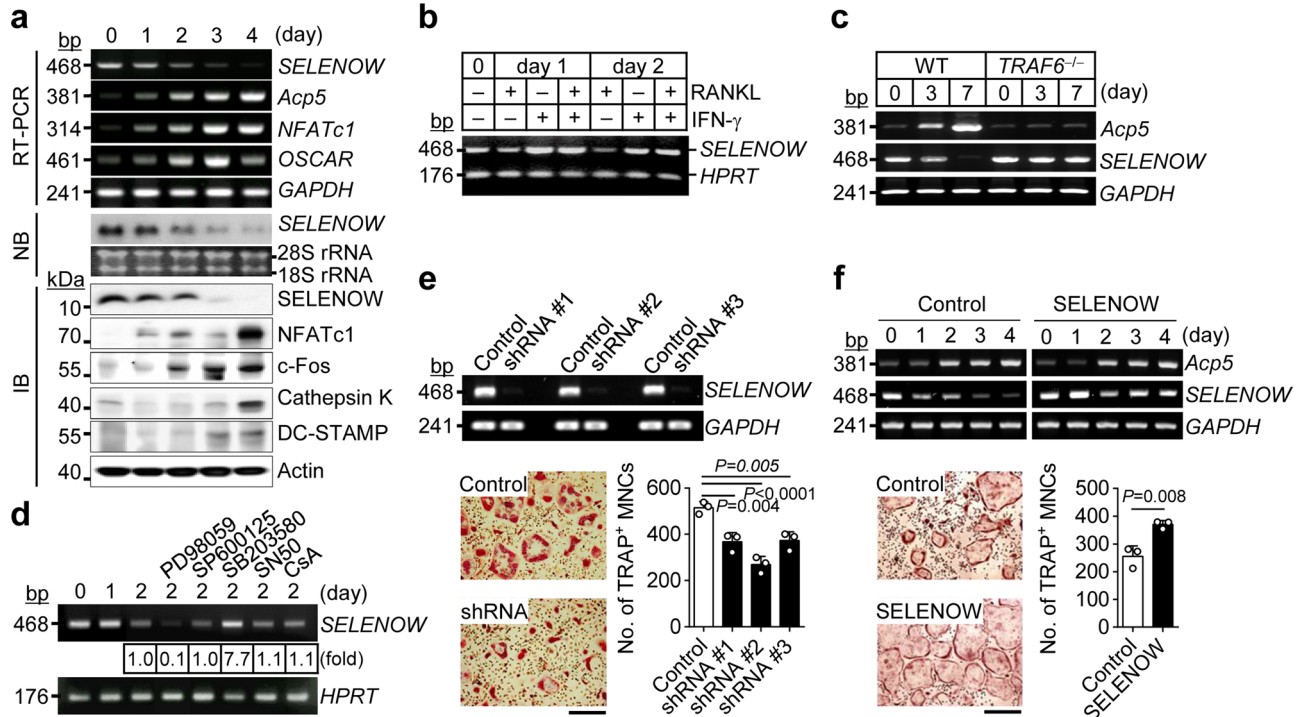

**Fig. 1 SELENOW positively regulates osteoclastogenesis. a** Downregulation of *SELENOW* during osteoclastogenesis. Osteoclast precursors were cultured with RANKL and M-CSF, and *SELENOW* gene expression was analysed by RT-PCR, northern blotting (NB), and immunoblotting (IB). **b, c** RANKL/RANK/ TRAF6 axis-dependent downregulation of *SELENOW*. Osteoclast precursors were pretreated with interferon-γ (IFN-γ; 150 U/ml), which degrades TRAF6, 30 min prior to RANKL stimulation. Osteoclast precursors treated with IFN-γ (**b**) and TRAF6-deficient osteoclast precursors (**c**) failed to induce RANKL-mediated *SELENOW* downregulation. **d** Up- and downregulation of *SELENOW* via ERK and p38 activation, respectively. Osteoclast precursors were pretreated with inhibitors of ERK (PD98059), JNK (SP600125), p38 (SB203580), NF-κB (SN50), and NFATc1 (cyclosporin A, CsA) for 30 min in the presence of M-CSF and then stimulated with RANKL for 2 days. The expression levels of *SELENOW* were analysed using RT-PCR. **e, f** Decreased and increased osteoclast formation following *SELENOW* knockdown (**e**) and overexpression (**f**), respectively. Osteoclast precursors infected with shRNA-mediated *SELENOW* gene-silencing lentivirus and *SELENOW*-overexpressing retrovirus were differentiated into osteoclasts and TRAP-positive multi-nucleated cells (TRAP + MNCs) with more than 3 nuclei were assessed (*n* = 3). Scale bars, 100 μm. Images are representative of three independent experiments. Data represent the mean ± SD of triplicate samples. Statistical significance was determined by Student's two-tailed *t*-test (**f**). One-way ANOVA was performed followed by Turkey's test (**e**). Source data are provided as a Source Data file.

differentiation and this effect differs from the inhibitory effects of other genes that are suppressed by RANKL. Moreover, these observations provide evidence that RANKL-dependent repression of SELENOW induces proper osteoclast differentiation.

**SELENOW deficiency and overexpression cause abnormalities in bone remodelling.** To investigate the physiological function of SELENOW, we developed *SELENOW*-deficient (*SELENOW*⁻/⁻) mice by generating transcription activator-like effector nucleases (TALENs) specific to exon 1 of *SELENOW*[37]. Also, we generated SELENOW; Lysozyme-Cre (SeW;LysM-Cre) mice showing *SELENOW*-deficient osteoclast precursors (macrophages). There was no SELENOW expression detected in any tissue of *SELENOW*⁻/⁻ mice or in osteoclast precursors derived from SeW;LysM-Cre mice (Supplementary Fig. 4a, b) or during RANKL-induced osteoclast differentiation of *SELENOW*⁻/⁻ mouse-derived osteoclast precursors (Supplementary Fig. 4c). Consistent with the suppression of osteoclastogenesis by shRNA-mediated *SELENOW* knockdown (Fig. 1e), osteoclast formation was markedly reduced in osteoclast precursors derived from *SELENOW*⁻/⁻ mice compared to that in osteoclast precursors from wild-type mice (Supplementary Fig. 4d), which was confirmed by the decreased levels of osteoclast markers, including NFATc1, tartrate-resistant acid phosphatase type (Acp) 5, and OSCAR (Supplementary Fig. 4e). *SELENOW*⁻/⁻ mice showed a slight increase in body weight compared to control mice

(Supplementary Fig. 5a), however, all other appearances including body size, tooth eruption, locomotor activity, and in bone-related parameters in the serum remained the same (Supplementary Fig. 5b–e). To further assess bone histology, we performed microcomputed tomography (μCT) analysis of trabecular bone in the proximal tibia and lumbar to analyse the physiological function of SELENOW in bone metabolism, and found that *SELENOW*⁻/⁻ mice had increased bone mass resulting from increases in trabecular bone volume, number, thickness, and mineral density and decreased trabecular bone separation (Fig. 2a and Supplementary Fig. 6a, c, d). There were fewer TRAP-positive osteoclasts on the surface of trabecular bone in *SELENOW*⁻/⁻ mice as compared to that in controls. However, the two groups had similar numbers of osteoblasts (Fig. 2b, c and Supplementary Fig. 6b). Moreover, bone histomorphometric characteristics of SeW;LysM-Cre mice were similar to those of *SELENOW*⁻/⁻ mice (Fig. 2d, e, f). SeW;LysM-Cre mice showed a decrease in BFR (Fig. 2g). These results indicate that the increased bone mass observed in *SELENOW*⁻/⁻ and SeW; LysM-Cre mice are caused by a reduction in osteoclast formation.

To further clarify the in vivo function of SELENOW in bone physiology, we generated transgenic mice in which *SELENOW* gene expression was controlled by the promoter of TRAP, which is highly expressed during osteoclast differentiation[38]. SELENOW was gradually upregulated during the differentiation of osteoclast precursors from the transgenic mice (Supplementary Fig. 7a). Ectopic expression of *SELENOW* in osteoclast precursors

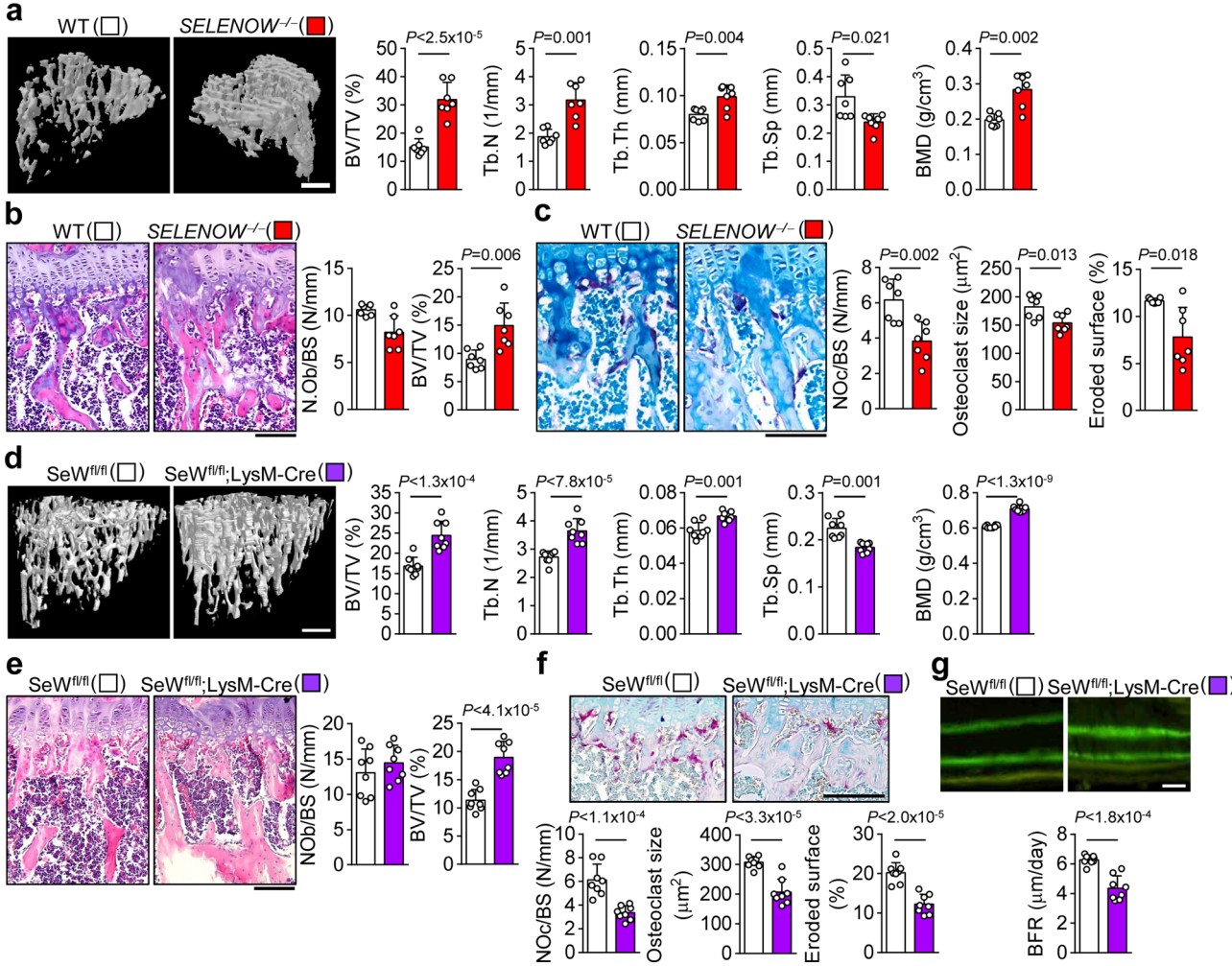

**Fig. 2 Increased bone mass phenotypes in mice with SELENOW deficiency. a** μCT analysis of proximal tibiae from wild-type (WT) male littermates and age/sex-matched *SELENOW*$^{-/-}$ mice at 10 weeks. BV/TV, trabecular bone volume per tissue volume; Tb.N, trabecular bone number; Tb.Th, trabecular thickness; Tb.Sp, trabecular separation; BMD, bone mineral density. Scale bar, 0.5 mm. **b, c** Reduced osteoclast formation on the trabecular bone surface of *SELENOW*$^{-/-}$ mice. H&E- and TRAP-stained sections of tibiae were used to detect osteoblasts (**b**) and osteoclasts (**c**), respectively. NOb/BS, number of osteoblasts per bone surface; NOc/BS, BV/TV, trabecular bone volume per tissue volume; number of osteoclasts per bone surface. In addition, osteoclast size and eroded bone surface were analysed from the TRAP-stained sections. Scale bar, 100 μm. **d** Increased bone mass phenotype in μCT analysis of proximal tibiae from WT male littermates (*SELENOW*$^{tm1c/tm1c}$; SeW$^{fl/fl}$) and age/sex-matched osteoclast-specific *SELENOW* knockout mice (*SELENOW*$^{tm/c/tm1c}$:LysM-Cre; SeW$^{fl/fl}$;LysM-Cre) at 10 weeks. Scale bar, 0.5 mm. **e** Analysis of NOb/BS, number of osteoblasts per bone surface, and BV/TV in H&E-stained sections. Scale bar, 100 μm. **f** Analysis of NOc/BS, number of osteoclasts per bone surface, osteoclast size and eroded bone surface from TRAP-stained sections. Scale bar, 100 μm. **g** Histomorphometric analysis of the tibia. BFR, bone formation rate. Scale bar, 10 μm. Data represent mean ± SD ($n = 7$ mice per group in **a–c**; $n = 8$ mice per group in **d–g**). Statistical significance was determined by Student's two-tailed *t*-test. Source data are provided as a Source Data file.

enhanced osteoclast formation as evidenced by the upregulation of osteoclastogenic genes (Supplementary Fig. 7b, c), which is consistent with the observation that retrovirus-induced *SELENOW* overexpression stimulated osteoclastogenesis (Fig. 1f). *SELENOW*-overexpressing transgenic mice exhibited a slight decrease in body weight compared to control littermates (Supplementary Fig. 8a), but other appearances, such as body size, tooth eruption, and locomotor activity, and in serum parameters related to bone metabolism remained the same (Supplementary Fig. 8b-e). In contrast to the high bone mass phenotype of *SELENOW*$^{-/-}$ mice, transgenic mice showed an osteoporotic phenotype in the trabecular bone of the tibia, calvaria, and lumbar, as determined using μCT scanning (Fig. 3a, b and Supplementary Fig. 9a, c, d). Histological analysis of bone tissue sections revealed that the transgenic mice had more multinucleated TRAP-positive osteoclasts than wild-type mice, whereas no differences were found in the numbers of osteoblasts

(Fig. 3c, d and Supplementary Fig. 9b). Additionally, TRAP staining of the whole calvaria, as well as cross sections, revealed enhanced osteoclast formation in transgenic mice (Fig. 3e, f). The area of the calvarial bone marrow cavity, which is an index of bone resorption activity[39], and the level of urinary deoxypyridinoline (DPD), a marker of bone resorption[40], were higher in transgenic as compared to that in wild-type mice (Fig. 3g, h). These results imply that the low bone mass in transgenic mice overexpressing *SELENOW* is caused by bone resorption due to enhanced osteoclast formation.

The expression of RANKL and osteoprotegerin (OPG), a decoy receptor for RANKL, which are produced by osteoblastic lineage cells and act as a positive and negative regulator for osteoclast differentiation, respectively[41], was no change in either the *SELENOW*$^{-/-}$ or the transgenic mice-derived osteoblasts (Supplementary Fig. 10a, b). To further assess the osteoclast-intrinsic function of SELENOW in bone remodelling, we assessed

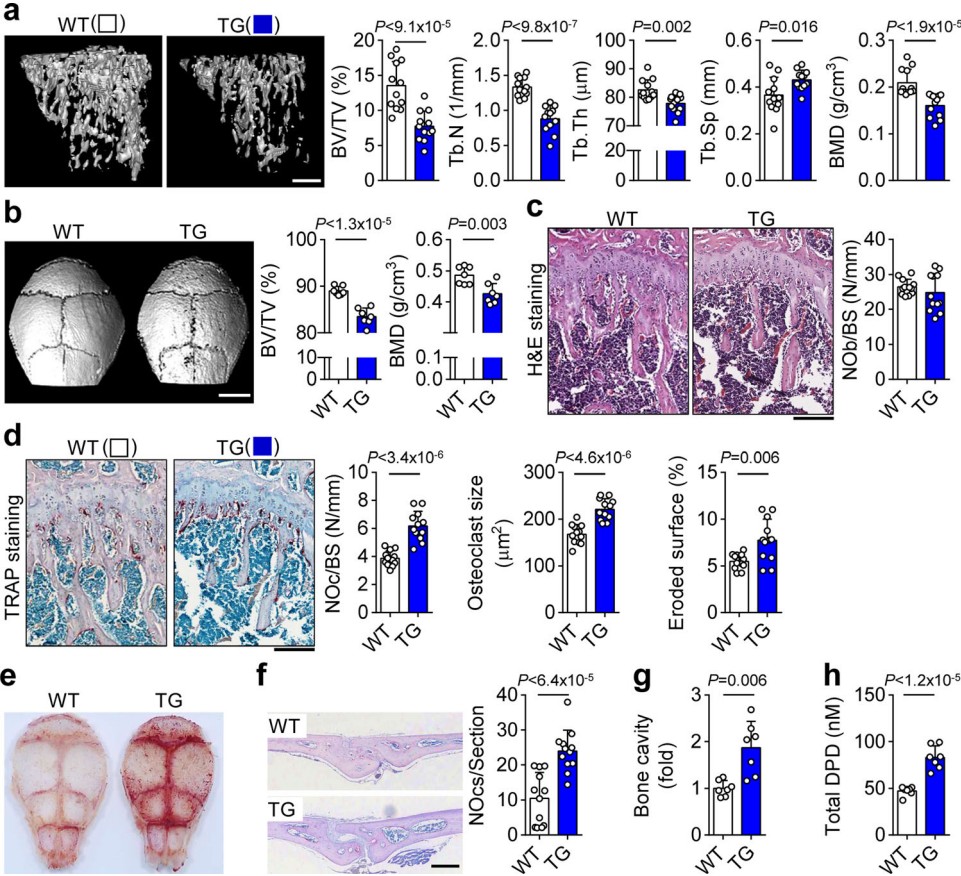

**Fig. 3 Osteoporotic phenotypes in mice with SELENOW overexpression. a** μCT analysis of proximal tibiae from wild-type (WT) male littermates and age/sex-matched transgenic (TG) mice at 10 weeks. Scale bar, 0.5 mm. **b** μCT images of calvaria and analysis of bone parameters [trabecular bone volume per tissue volume (BV/TV) and BMD]. scale bar, 3 mm. **c**, **d** Increased osteoclast formation on the trabecular bone surface of TG mice. Analysis of NOb/BS, number of osteoblasts per bone surface, from H&E-stained sections (**c**). Analysis of NOc/BS, number of osteoclasts per bone surface, osteoclast size and eroded bone surface from TRAP-stained sections (**d**). Scale bar, 100 μm. **e–g** Whole calvaria (**e**) and cross-sections (**f**) were stained with TRAP. The number of TRAP + osteoclasts and the calvarial marrow cavity area (**g**), which reflects the degree of osteoporosis, were measured in whole sections. Scale bar, 1 mm. **h** The level of urinary DPD, a marker of osteoporosis, was measured by enzyme immunoassay. Data represent mean ± SD ($n = 12$ mice per group in **a**, **c**, **d** and **f**, and $n = 7$ mice per group in **b** and **e–h**). Statistical significance was determined by Student's two-tailed $t$-test. Source data are provided as a Source Data file.

osteoclast formation using a co-culture system of calvarial osteoblasts and bone marrow osteoclast precursors prepared from $SELENOW^{-/-}$ and transgenic mice. Regardless of expression levels of SELENOW in osteoblasts, osteoclast precursors derived from $SELENOW^{-/-}$ and transgenic mice displayed decreased and increased osteoclast differentiation, respectively (Supplementary Fig. 10c, d). The combined observations from in vitro and in vivo data suggest that SELENOW-regulated bone remodelling results from a cell-autonomous effect on osteoclast formation. Thus, the regulation of SELENOW expression allows proper osteoclast formation and may be important for normal bone turnover.

**SELENOW stimulates osteoclastogenesis via activation of NF-κB and NFATc1.** To clarify the molecular mechanism by which SELENOW regulates osteoclastogenic factors, we examined the activation of signalling cascades in immediate or delayed response to RANKL. We found that SELENOW triggered NF-κB activation—as evidenced by increased degradation of inhibitor of NF-κB (IκBα)—but did not affect MAPK (ERK, p38, and JNK) signalling (Supplementary Fig. 11). In addition, SELENOW promoted the nuclear translocation of cytosolic NF-κB and NFATc1 without altering the expression levels of these proteins (Fig. 4a). To confirm the

stimulatory effect of SELENOW on NF-κB- and NFATc1-dependent transcriptional activity, we assessed the activity of promoters with a binding site for NF-κB or NFATc1. A luciferase reporter assay showed that RANKL-dependent NF-κB- and NFATc1-induced promoter activity was increased in SELENOW-overexpressing as compared to control cells (Fig. 4b). In contrast, there was no change in the activity of a promoter harbouring an AP-1-binding site in SELENOW-overexpressing cells. His-tag pull-down and immunoprecipitation (IP) assays revealed that SELENOW interacted with NF-κB and NFATc1 but not with AP-1 (Fig. 4c and Supplementary Fig. 12). To identify the SELENOW residue responsible for the interaction with NF-κB or NFATc1, we developed SELENOW mutants, replacing SeCys-13 with either cysteine or serine residues. SELENOW mutant forms did not interact with NF-κB or NFATc1 (Fig. 4d). These results indicate that the SeCys-13 SELENOW residue is necessary for the interaction with either NF-κB or NFATc1 and may be involved in regulating their activity. To further analyse the interaction between SELENOW and NF-κB or NFATc1, chromatin (Ch)IP was performed using an antibody specific to SELENOW or control IgG; immunoprecipitated DNA was PCR amplified with primers recognising promoters containing an NF-κB- or NFATc1-binding site[42,43]. A specific PCR product was detected in DNA from cells exhibiting relatively high levels of SELENOW—including undifferentiated osteoclast

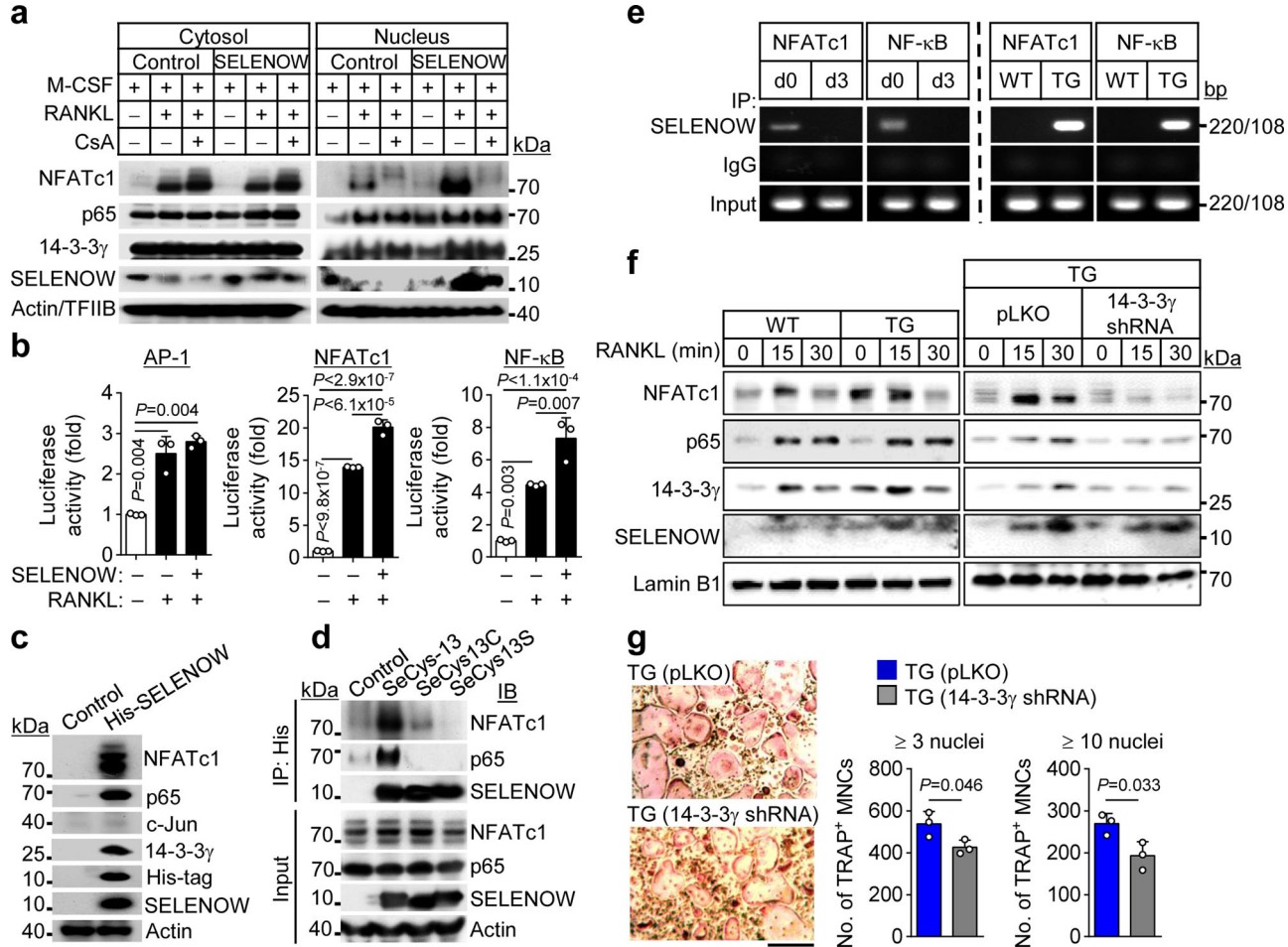

**Fig. 4 SELENOW regulates osteoclastogenic gene expression. a** Osteoclastogenic transcription factors and SELENOW co-translocate into the nucleus. Osteoclast precursors infected with *SELENOW*-harbouring retrovirus were cultured with M-CSF and RANKL for 2 days. After cells were exposing to RANKL-free condition for 3 h and treated without or with an inhibitor of NFATc1 (cyclosporin A, CsA), cells were stimulated with RANKL for 20 min. Cytosolic and nuclear proteins were fractionated and NF-κB, NFATc1, and SELENOW levels were determined by immunoblotting. **b** Luciferase reporter assay. RAW264.7 cells were transfected with AP-1-, NF-κB-, and NFATc1-luciferase reporter or pcDNA3.1-His-tagged SELENOW (SeCys-13) vector. Cells were stimulated with RANKL for 24 h and luciferase activity was measured (*n* = 3). **c, d** SELENOW interacts with NF-κB and NFATc1. Cytosolic extracts from HEK 293 T cells expressing a His-tagged SELENOW (SeCys-13) were pulled down with an anti-His-Tag antibody (**c**). Also, cytosolic extracts from HEK 293T cells with a His-tagged wild-type SELENOW (SeCys-13) and His-tagged SELENOW mutants in which SeCys-13 was replaced by cysteine (SeCys13C) or serine (SeCys13S) were immunoprecipitated (IP) with anti-His-Tag antibody and then immunoblotted (IB) with the indicated antibodies (**d**). **e** ChIP assay. Osteoclast precursors were cultured with M-CSF alone (d0) or with M-CSF and RANKL for 3 days (d3; left panels). Also, osteoclast precursors from wild-type (WT) and *SELENOW*-overexpressing transgenic (TG) mice were cultured with M-CSF and RANKL for 3 days (right panels). Following immunoprecipitation (IP) of chromatin with anti-SELENOW antibody, ChIP assay was performed to detect the promoter for NF-κB- or NFATc1-binding sites. **f, g** 14-3-3γ mediates nuclear translocation of NFATc1, NF-κB, and SELENOW, and osteoclast differentiation. After osteoclast precursors from WT and TG mice were cultured with M-CSF and RANKL for 2 days to induce pre-osteoclasts, the cells were exposed to M-CSF- and RANKL-free condition for 3 h and were stimulated with RANKL for indicated times (**f**; left panel). In addition, this was performed in TG mice-derived pre-osteoclasts transduced with control lentivirus (pLKO) or 14-3-3γ-targeted shRNA-harbouring lentivirus (**f**; right panel). Nuclear proteins were fractionated and subjected to immunoblotting. Osteoclast precursors from TG mice were infected with shRNA-mediated 14-3-3γ gene-silencing lentivirus and differentiated into osteoclasts (*n* = 3). TRAP + MNCs with more than 3 or 10 nuclei were assessed (**g**). Scale bar, 100 μm. Data represent the mean ± SD of triplicate samples. Statistical significance was determined by Student's two-tailed *t*-test (**g**). One-way ANOVA was performed followed by Turkey's test (**b**). Images are representative of three independent experiments. Source data are provided as a Source Data file.

precursors (Fig. 4e; left panels) and differentiated osteoclasts formed by *SELENOW*-overexpressing transgenic mouse osteoclast precursors (Fig. 4e; right panels)—but not in cells with low *SELENOW* expression. We further validated the SELENOW-dependent activation of NFATc1 and NF-κB. RANKL treatment resulted in an increased and a decreased nuclear import of NFATc1 and NF-κB in pre-osteoclasts from *SELENOW*-transgenic mice and in pre-osteoclasts from *SELENOW*−/− mice, respectively (Fig. 4f; left panel and Supplementary Fig. 13a). Interestingly, nuclear translocation of 14-3-3γ also increased in pre-osteoclasts from

*SELENOW*-transgenic mice and decreased in pre-osteoclasts from *SELENOW*−/− mice. Considering that 14-3-3 is known to regulate the subcellular localisation of its interacting partners[44] and bind to SELENOW[45,46], we next investigated the role of 14-3-3γ in SELENOW-induced activation of NFATc1 and NF-κB and osteoclast formation, using a lentivirus carrying an shRNA (Supplementary Fig. 13b, c). 14-3-3γ knockdown significantly suppressed nuclear translocation of NFATc1, NF-κB, and SELENOW (Fig. 4f; right panel, Supplementary Fig. 13d), leading to decreased osteoclast formation (Fig. 4g). These results indicate that SELENOW induces

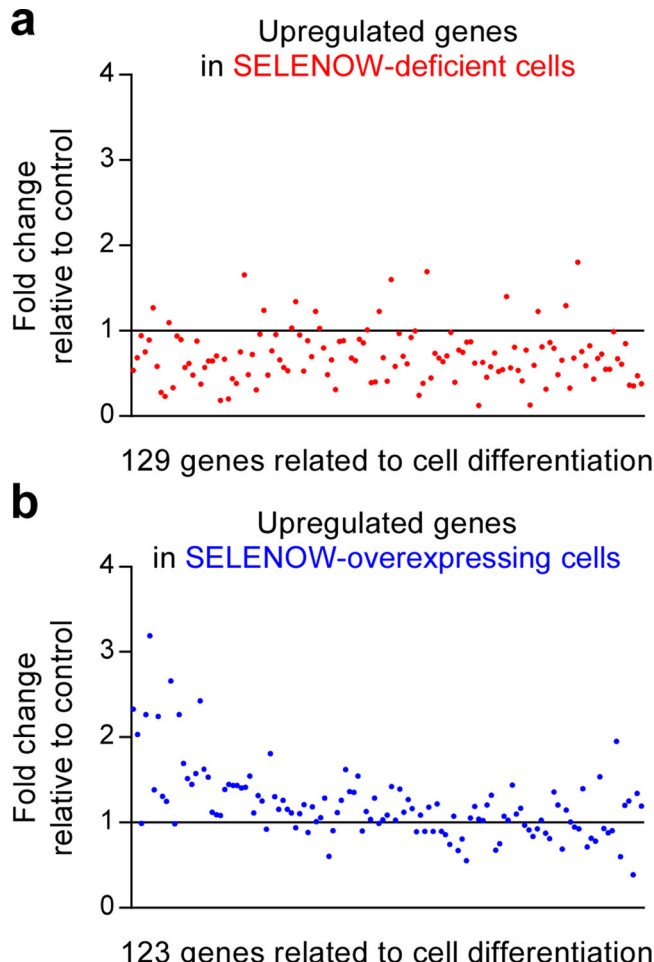

**Fig. 5 SELENOW upregulates osteoclast differentiation-related genes. a, b** Analysis of RNA-sequencing transcriptomic data. Osteoclast precursors from *SELENOW*$^{-/-}$ and *SELENOW*-transgenic mice and corresponding wild-type littermates were differentiated into osteoclasts in the presence of M-CSF and RANKL for 3 days. The relative fold change in the levels of genes in *SELENOW*$^{-/-}$ (**a**) and *SELENOW*-overexpressing osteoclasts (**b**) was determined. Source data are provided as a Source Data file.

osteoclast differentiation by promoting the nuclear translocation of NF-κB/SELENOW or NFATc1/SELENOW cytosolic complexes via 14-3-3γ, and that this effect is progressively attenuated by RANKL to prevent excess osteoclast formation during osteoclastogenesis.

Our data suggest that dysregulation of SELENOW expression results in abnormal osteoclast formation and bone remodelling and perturbation of RANKL-dependent, NF-κB and NFATc1-induced osteoclastogenic signalling. We evaluated the expression of cell differentiation-related genes controlled by SELENOW in osteoclast precursors derived from *SELENOW*$^{-/-}$ C57BL6 and *SELENOW*-overexpressing FVB3 mice or wild-type C57BL6 and FVB3 mice that were differentiated into osteoclasts in the presence of RANKL for 3 days by RNA-sequencing transcriptome analysis. Genes that were upregulated before (day 0) and after (day 3) differentiation of wild-type osteoclast precursors into osteoclasts were compared between *SELENOW*$^{-/-}$ and *SELE-NOW*-overexpressing cells relative to their respective wild-type cells (Supplementary Fig. 14a, b, d, e). When the fold-induction of cell differentiation-related upregulated genes in wild-type cells was set to 1, most of the upregulated genes in *SELENOW*$^{-/-}$ cells had lower expression than those in wild-type cells (Fig. 5a), whereas a large portion of the upregulated genes in

*SELENOW*-overexpressing cells had higher expression than those in wild-type cells (Fig. 5b). A large proportion of genes showing 1.5-fold decrease or increase in *SELENOW*$^{-/-}$ or *SELENOW*-overexpressing cells relative to those in the wild-type were identified as osteoclastogenesis-promoting factors, including *Itgav* (encoding integrin αv), *Itgb3*, DC-STAMP, and NFATc1, and were predominantly altered in *SELENOW*$^{-/-}$ rather than *SELENOW*-overexpressing cells (Supplementary Fig. 15a). These results indicate that complete loss of *SELENOW* leads to more severe dysregulation of RANKL-induced upregulation of osteoclastogenic differentiation-related genes than its constitutive expression.

We next analysed genes that were downregulated by RANKL, which are considered as negative regulators of osteoclastogenic differentiation (Supplementary Fig. 14c, f and Supplementary Fig. 15b). The levels of differentiation-associated downregulated genes were decreased and increased in *SELENOW*$^{-/-}$ and *SELENOW*-overexpressing cells, respectively, relative to the corresponding wild-type cells (Supplementary Fig. 15c). Furthermore, NF-κB- and NFATc1-dependent osteoclastogenic genes [e.g. *Prdm1* (encoding PR/SET domain 1, also termed *Blimp1*), *Itgav*, *Itgb3*, and *CAV1* (encoding caveolin-1)][47–50] were among the differentiation-related genes in *SELENOW*$^{-/-}$ and *SELE-NOW*-overexpressing cells whose expression was altered in response to RANKL (Supplementary Fig. 15a), indicating that SELENOW modulates RANKL-induced osteoclastogenic genes activated by NF-κB and NFATc1.

**Constitutive *SELENOW* expression stimulates pre-osteoclast fusion and osteoclastic bone resorption.** Our in vitro and in vivo results showed that ectopic expression of *SELENOW* enhances osteoclast formation. This prompted us to investigate the effect of constitutive *SELENOW* expression during osteoclast differentiation on osteoclast metabolism, including the mechanism by which SELENOW facilitates osteoclast differentiation and whether SELENOW can affect the bone-resorptive activity of mature osteoclasts.

To evaluate pre-osteoclast fusion—a critical step in the formation of multi-nucleated osteoclasts[51]—we prepared pre-osteoclasts by culturing osteoclast precursors in a medium containing M-CSF and RANKL for 2 days. The rate of cell-cell fusion was increased in *SELENOW*-overexpressing mononuclear pre-osteoclasts as compared to control cells (Fig. 6a), which was associated with increased mRNA expression of fusion-associated genes including *Itgav* and *Itgb3* (Supplementary Fig. 15a; right panel). Accordingly, cell-cell fusion was reduced in *SELENOW*$^{-/-}$ pre-osteoclasts relative to control cells (Fig. 6b), with a corresponding decrease in *Itgav* and *Itgb3* and other fusion-related factors such as *DC-STAMP* and *osteoclast-STAMP* (Supplementary Fig. 15a; left panel). We also observed that SELENOW stimulated osteoclastic bone-resorptive activity (Fig. 6c, d) and suppressed the apoptosis of mature osteoclasts, as evidenced by an increase in the number of mature osteoclasts with a full actin ring and decreases in caspase-9 and -3 activities (Fig. 6e, f). Previous results obtained by our group and others have shown that cellular redox status is shifted towards oxidation during osteoclast differentiation[52,53]; meanwhile, SELENOW is thought to function as an antioxidant[54]. To investigate this possibility, we evaluated the change in redox status following osteoclast maturation and examined whether this—and consequently, osteoclast lifespan—is modulated by SELENOW. Consistent with previous reports, redox status was shifted toward an oxidised state after mature osteoclast formation (Fig. 6g, h). In contrast, mature osteoclasts overexpressing *SELENOW* showed a marked increase in reduction capacity as compared to control cells (Fig. 6g). We speculated that the prolonged survival of

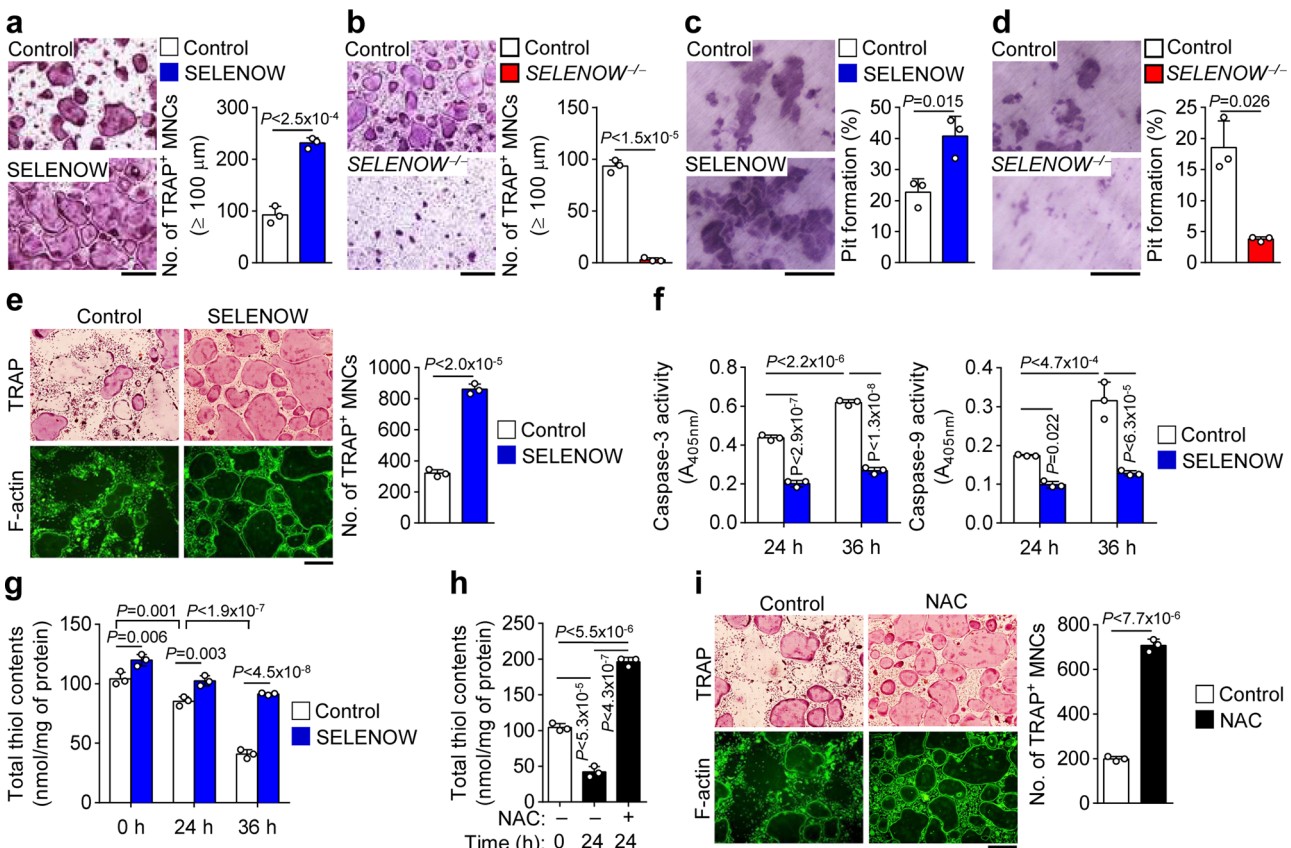

**Fig. 6 Constitutive expression of SELENOW leads to excess pre-osteoclast fusion and osteoclastic bone resorption. a, b** Induction of pre-osteoclast fusion and osteoclastic bone resorption by SELENOW. Fusion assay in pre-osteoclasts from wild-type and *SELENOW*-overexpressing transgenic mice (**a**) or *SELENOW*−/− mice (**b**). Osteoclast precursors were treated with M-CSF and RANKL for 2 days to form pre-osteoclasts following fusion assay. Osteoclast fusion rate was determined by counting TRAP + MNCs with a diameter ≥100 μm (n = 3). **c, d** Pit formation. Osteoclast precursors prepared from wild-type and *SELENOW*-overexpressing transgenic mice (**c**) or *SELENOW*−/− mice (**d**) were differentiated into osteoclasts for 4 days. After mature osteoclasts were detached from the culture dish and seeded on dentine slice, cells were further cultured with M-CSF and RANKL for 2 days to allow bone resorption. Pit formation by osteoclasts is expressed as a percentage of the resorbed area on the bone slice surface (n = 3). **e** Anti-apoptotic effect of SELENOW. Mature osteoclasts were transduced with *SELENOW*-overexpressing retrovirus and cell survival was assessed 2 days later by staining with TRAP (upper panels) or FITC-labelled phalloidin (lower panels) to detect TRAP + osteoclasts with a full actin ring (n = 3). **f** Caspase activity was assessed at indicated times after mature osteoclasts were cultured as in (**e**, n = 3). **g** Increase in the cellular redox status by SELENOW. Osteoclasts were transduced with *SELENOW*-overexpressing retrovirus and total thiol content was assessed at indicated times (n = 3). **h** Increase in the cellular redox status by NAC. After treatment with 4 mM NAC for 24 h or no treatment, cytosolic extracts of mature osteoclasts were prepared and assayed for free thiol level (n = 3). **i** Increased mature osteoclast survival by NAC (n = 3). Osteoclasts were treated as described in **h** and then stained as in **e**. Scale bars, 100 μm. Data represent mean ± SD of triplicate samples. Statistical significance was determined by Student's two-tailed *t*-test (**a–e**, **i**). One-way ANOVA was performed followed by Turkey's test (**f–h**). Source data are provided as a Source Data file.

osteoclasts by *SELENOW* overexpression was due in part to their high reduction potential resulting in enhanced bone resorption. The relationship between osteoclast survival and redox status was confirmed by the finding that osteoclast lifespan was extended by treatment with the antioxidant N-acetylcysteine, which enhanced osteoclast reduction capacity (Fig. 6i). Collectively, our results demonstrate that dysregulation of RANKL-mediated inhibition of *SELENOW*, which is constitutively expressed during osteoclast differentiation, promotes osteoclast maturation by stimulating pre-osteoclast fusion and leads to excessive bone resorption as a result of increased osteoclast longevity.

## Discussion

It has been previously reported that osteoporotic bone loss occurs when osteoblast-mediated bone formation uncouples from osteoclast-induced bone resorption[55]. An absolute increase in the rate of bone resorption with an insufficient increase in the rate of bone formation results in bone loss. Hyperactive osteoclast

activity could cause unbalanced bone remodelling and metabolic bone diseases such as osteoporosis, metastatic cancers, and rheumatic arthritis[41,56]. RANKL is essential for osteoclast differentiation and can induce positive osteoclastogenic factors including NFATc1, OSCAR, NF-κB, Atp6v0d2, DC-STAMP, and c-Fos. RANKL is also involved in a negative-feedback auto-regulatory mechanism thus preventing overactive osteoclast activity and helps to maintain bone homoeostasis[57,58]. Two negative regulators of osteoclastogenesis, glutathione peroxidase 1 and interferon-β, were induced by RANKL during osteoclast differentiation. Oxidative stress mediated by reactive oxygen species (ROS), such as hydrogen peroxide and superoxide, has been reported to promote RANKL-induced osteoclast differentiation[59]. Glutathione peroxidase 1, the first identified mammalian selenoprotein containing a SeCys at their active sites and is expressed during RANKL-induced osteoclast differentiation, acts as a scavenger of hydrogen peroxide[58]. Overexpression of glutathione peroxidase 1 in osteoclasts was shown to prevent RANKL-induced osteoclast differentiation. It has been also reported that

RANKL stimulates gene expression of interferon-β. Interferon-β suppressed the RANKL-induced expression of c-Fos, an important transcription factor for osteoclastogenesis, functioning as a negative-feedback regulator to inhibit osteoclast differentiation[57]. RANKL signalling induces the expression of two negative regulators, Gpx-1 and interferon-β, which stimulates a negative-feedback loop that contributes to maintain the normal bone mass. In this study, we showed that SELENOW, a positive regulator of osteoclastogenesis, is gradually and significantly repressed during osteoclast differentiation, demonstrating that SELENOW plays a physiologically critical role in the negative-feedback regulation of osteoclastogenesis and prevents hyperactive osteoclast activity and excessive bone resorption. In conclusion, RANKL is involved in a negative-feedback auto-regulatory mechanism for the maintenance of bone homoeostasis through both upregulation of negative regulators and downregulation of positive regulator during osteoclast differentiation.

The results of this study indicate that TRAF6-independent ERK activation mediates RANKL-induced SELENOW upregulation while TRAF6-dependent p38 activation facilitates RANKL-induced SELENOW downregulation. This raises the question of how SELENOW can be predominantly downregulated during osteoclast differentiation when RANKL-activated ERK and p38—which promote and inhibit SELENOW expression, respectively—are both present. We propose that this is possible because RANKL stimulation induces stronger and longer-lasting activation of p38 than M-CSF stimulation (Supplementary Fig. 2d). Our results also showed that the constitutive expression of RANKL-repressed SELENOW promotes osteoclastogenesis. This suggests a unique regulatory circuit for SELENOW in osteoclastogenesis that is distinct from those of previously characterised RANKL-repressed genes [e.g. Id2, MafB, IRF8, Bcl6, and Lhx (encoding LIM homeobox proteins)], whose constitutive expression inhibits osteoclast differentiation[11–15]. It is likely that an adequate level of SELENOW in osteoclast precursors allows initiation of osteoclast differentiation, while its progressive disappearance during differentiation prevents excessive osteoclast formation. A specific interaction between SELENOW and 14-3-3 protein is known to involve the conserved redox motif CysXXSeCys present in a SELENOW exposed loop, and an exposed Cys residue in the C-terminal domain of 14-3-3 protein[21]. The binding of SELENOW to 14-3-3 protein was significantly enhanced under oxidative stress conditions[46], indicating that the intracellular redox status regulates SELENOW-14-3-3 protein interactions, and modulates 14-3-3 protein functional activity. Here, we observed that the SELENOW SeCys-13 residue is critical for the interaction with its targets, NFATc1 and NF-κB, and that SELENOW leads to an increase in cell reduction capacity. Also, we previously highlighted that this residue, SeCys-13, is crucial for SELENOW antioxidant activity[54]. Altogether, these findings suggest that SELENOW has the potential to enhance osteoclast differentiation via direct interaction with NFATc1 and NF-κB and their subsequent activation, but also indirectly, as a reducing agent.

14-3-3 proteins have been reported to participate in various cellular pathways such as cell cycle, differentiation, survival, and apoptosis by altering the subcellular localisation of their binding partners[44]. In particular, 14-3-3 was shown to enhance nuclear localisation of thioredoxin-like protein 2, checkpoint kinase 1, telomerase reverse transcriptase, and T-cell protein tyrosine phosphatase[60–63]. We here demonstrated that 14-3-3γ mediates nuclear translocation of NF-κB and NFATc1 induced by SELENOW, contributing to osteoclastogenesis. Based on the coordination between SELENOW and its interacting partners from our pull-down and immunoprecipitation assay (Fig. 4c and Supplementary Fig. 12), we found that SELENOW interacts with NF-κB,

NFATc1, and 14-3-3γ. 14-3-3γ formed a complex with SELENOW and NFATc1, but not NF-κB. We also observed that NFATc1 interacts with SELENOW, NF-κB, and 14-3-3γ, and NF-κB forms a complex with SELENOW and NFATc1, but not 14-3-3γ. These results suggest that NFATc1 interacts directly with SELENOW and 14-3-3γ, and NF-κB interacts directly with SELENOW and indirectly with 14-3-3γ. Further study on the interaction of SELENOW with NFATc1 and NF-κB through 14-3-3 protein will extend our knowledge on the regulation of osteoclastogenesis by SELENOW.

Previous studies have reported that bone metastases in some cancers, such as breast, prostate, and multiple myeloma, lead to the release of osteoclast-activating factors (e.g. $\beta_2$-microglobulin, IL-1β, and TNF-α) from myeloma cells, T cells, marrow stromal cells, and monocytes, thereby promoting osteoclastogenesis and osteoclastic bone resorption[64–66]. Ria et al.[67] also reported that bone marrow endothelial cells in active multiple myeloma patients with osteolytic lesions had notably induced SELENOW expression, which may influence disease progression. Since SELENOW may exert different effects depending on the disease type and severity, further studies are needed to investigate the specific role of SELENOW in various bone defects, including bone metastatic and osteoporosis-related cancers, for the development of anti-osteoporotic agents using SELENOW.

Oxidative stress mediated by ROS has been shown to be deleterious to normal bone physiology. A number of antioxidant selenoenzymes are known to be involved in both induction of osteoblast differentiation and inhibition of osteoclast differentiation by removing intracellular ROS levels, thus improving bone health[29,68]. The deficiency of selenium and selenoproteins in bone could potentially have negative effects on bone metabolism due to the failure to suppress oxidative stress in both osteoblasts and osteoclasts[24,69]. However, we showed that whole-body SELENOW-deficient and osteoclast-specific SELENOW-deficient mice exhibited increased bone mass phenotype in which bone resorption was impaired due to osteoclast dysfunction, with the activity of osteoblasts remaining unchanged. On the contrary to the negative impacts of selenium insufficiency on bone homoeostasis due to decreased osteoblast and increased osteoclast function, increased bone mass by SELENOW deficiency may have resulted from the attenuated osteoclastic activity with no alteration in osteoblastic activity.

Our results indicate that the anti-apoptotic effect of SELENOW in osteoclasts may be due to an increase in the cellular reduction status. It is presumed that abnormal, constitutive expression of SELENOW can prolong osteoclast survival by creating a reducing environment in cells, resulting in excess bone resorption. However, in normal osteoclastogenesis, the stimulatory action of SELENOW may gradually be suppressed by RANKL, thereby regulating osteoclast differentiation and function. Collectively, our findings highlight an unusual regulatory circuit governing osteoclast differentiation and provide novel insight into the physiological roles of selenoproteins in bone metabolism.

## Methods
**Osteoclast and osteoblast differentiation.** Bone marrow-derived macrophages, which are osteoclast precursors, were obtained from the femur and tibia of 6-week-old male C57BL/6J mice (Central lab animal, Seoul, Korea) as previously reported[52], unless otherwise indicated. TRAF6$^{-/-}$ osteoclast precursors were prepared from liver-derived macrophages obtained from day 14.5 to 16.5 TRAF6$^{-/-}$ C57BL/6J embryos[70]. For osteoclast differentiation, osteoclast precursors ($2 \times 10^4$ cells/well in a 48-well plate) were cultured in α-Minimal Essential Medium (α-MEM) with M-CSF (30 ng/ml) and RANKL (100 ng/ml) for 4 days with the medium changed after 2 days. To assess osteoclast differentiation, cells were fixed with 3.7% (v/v) formaldehyde in phosphate-buffered saline (PBS) for 10 min and stained for TRAP with a leukocyte acid phosphatase staining kit (387 A; Sigma-Aldrich, St. Louis, MO, USA). TRAP-positive multi-nucleated cells (TRAP + MNCs) with more than three nuclei were counted under a light microscope. To

induce osteoblast differentiation, primary osteoblast precursors were isolated from the calvarial bone of newborn (1–2 days) C57BL6 mice (Central lab animal, Seoul, Korea) by sequential digestion with dispase II and collagenase type IA. Osteoblast precursors ($1.5 \times 10^6$ cells/well in a 6-well plate) were infected with lentivirus or retrovirus for *SELENOW* gene-silencing and overexpression, respectively. After selecting with puromycin (2 µg/ml) for 2 days, puromycin-resistant cells were re-seeded in a 48-well plate at a density of $4 \times 10^4$ cells/well and cultured in osteogenic medium (α-MEM with 100 µg/ml ascorbic acid and 10 mM β-glycerol phosphate) for 8 days with the medium changed every 2 days. After 4 or 8 days of culture, the cells were fixed in 95% ethanol for 30 min and stained with 1% Alizarin Red S solution (pH 4.2; Sigma-Aldrich) for 30 min at 37 °C. To assess matrix calcification, the stain was solubilized with 10% cetylpyridinium chloride (pH 7.0) by shaking for 15 min and the absorbance of the released Alizarin Red S was measured at 570 nm. Bone marrow-derived primary osteoclast precursors ($1 \times 10^5$ cells/well in a 48-well plate) were co-cultured with calvarial primary osteoblast cells ($1 \times 10^4$ cells/well in the presence of 20 nM 1α,25-dyhydroxy vitamin D3 and 1 mM prostaglandin E2) for 12 days; the medium was changed after 2 days. The extent of osteoclast formation was determined by counting TRAP + MNCs with more than three nuclei.

**GeneChip analysis.** Osteoclast precursors were cultured with M-CSF (30 ng/ml) and RANKL (100 ng/ml) for the indicated times. Total RNA (550 ng) was used to synthesise cDNA by reverse transcription. Biotinylated cRNA was generated using the Ambion Illumina RNA Amplification kit (Applied Biosystems) and hybridised to the mouse-6 expression head array (Illumina) according to the manufacturer's instructions. The level of representative genes showing a 3-fold increase or decrease change is depicted as fold-induction or repression relative to control osteoclast precursors treated with M-CSF alone.

**Cell fusion and bone pit formation assays.** In the cell fusion assay, pre-osteoclasts treated with M-CSF (30 ng/ml) and RANKL (100 ng/ml) for 2 days were seeded at a density of $1 \times 10^5$ cells/well in a 48-well plate until 100% confluence, and then cultured with M-CSF and RANKL for 2 days. After TRAP staining, the number of TRAP + MNCs with a diameter greater than 100 µm was counted as a measure of osteoclast fusion. For the pit formation assay, osteoclast precursors with differential SELENOW expression were differentiated into osteoclasts for 4 days; the cells were then detached from the culture dish and seeded on a dentine slice (IDS Ltd., Tyne & Wear, UK) at a density of $1 \times 10^4$ cells/well in a 96-well plate, and cultured with M-CSF and RANKL for 2 days to allow bone resorption. After removing adherent cells from the dentine slices by ultrasonication, the area of resorbed pits stained with haematoxylin (Sigma-Aldrich) was photographed under a light microscope and was measured using Image-Pro Plus v.6.0 software (Media Cybernetics, Silver Spring, MD, USA).

**Viral infection.** For *SELENOW* gene-silencing, osteoclast precursors were transduced with lentiviral particles carrying *SELENOW*-targeted shRNA (TRCN0000292860, TRCN0000292859, and TRCN0000298060; Sigma-Aldrich) and 14-3-3γ-targeted shRNA (TRCN0000337504; Sigma-Aldrich), and incubated overnight. To induce efficient *SELENOW* knockdown, virus-infected cells were selected by stepwise increases in puromycin concentration from 0.25 to 2 µg/ml over 4 days in the presence of M-CSF (30 ng/ml). For ectopic expression of *SELENOW*, a 510-bp fragment produced by PCR using primers with a *Bgl*II site (sense, 5′-CGCGAGATCTATGGCGCTCGCCGTTCGAGTC-3′ and antisense, 5′-GGCGAGATCTTTTCAGAGAGAGAGGTGGGGAA-3′) was cloned into the *Bam*HI site of the pMX-puro retroviral vector that was transfected to Plat-E cells as previously described[11] using Lipofectamine 2000 (Invitrogen, Carlsbad, CA, USA). The retrovirus was collected from the culture medium 48 h later. Osteoclast precursors were infected with retrovirus in a medium containing 60 ng/ml M-CSF in the presence of 10 µg/ml polybrene for 8 h, and then selected with puromycin (2 µg/ml) for 2 days in the presence of M-CSF (120 ng/ml). The efficiency of *SELENOW* knockdown and overexpression using lenti- and retroviruses, respectively, was confirmed by reverse transcription (RT)-PCR. Virus-infected osteoclast precursors were used for experiments, including osteoclast differentiation.

**Genetically modified mice.** $SELENOW^{-/-}$ C57BL6 mice were generated using TALENs specific to exon 1 of *SELENOW* as previously described[37]. Genomic DNA was isolated from mouse tail biopsies and used for PCR genotyping with Maxima Hot Start Green PCR Master Mix (Thermo Fisher Scientific, Waltham, MA, USA) and the following primers: sense, 5′-CGTAGCTCTGCCCACTCTCCAC-3′ and antisense, 5′-AGCAGGAAAAGGGGGAACTG-3′. The wild-type and knockout alleles were represented by 213- and 188-bp PCR products, respectively. *SELENOW* deficiency in various tissues and during osteoclast differentiation was confirmed by immunoblotting using an antibody against SELENOW (Novus Biologicals, Littleton, CO, USA). To produce the conditional *SELENOW*-deficient mice using osteoclast precursor (monocyte/macrophage), mice carrying the floxed conditional $SELENOW^{tm1c}$ allele in the C57BL/6 background were purchased from the European Mouse Mutant Archive (EMMA, ID EM:09166, Wellcome Trust Sanger Institute, Prague, Czech Republic). The monocytic cell-specific Lysozyme M (LysM)-Cre mice (Stock Number: 004781) in the C57BL/6 background were obtained from Jackson Laboratory (Bar Harbor, USA). The conditional

$SELENOW^{tm1c/tm1c}$ mice were bred with LysM-Cre mice in order to generate mice deficient for *SELENOW* in the hematopoietic myeloid-osteoclast lineage ($SELENOW^{tm/c/tm1c}$;LysM-Cre). $SELENOW^{tm1c/tm1c}$ mice were used as wild-type littermates. To generate *SELENOW*-overexpressing transgenic mice, the amplified 510-bp PCR fragment with the SECIS necessary for *SELENOW* gene expression was digested with *Bgl*II and then cloned into the *Bam*HI site of the pBluescript vector containing the promoter of TRAP—which is highly expressed during osteoclast differentiation—and a poly A site. The vector was injected into the pronucleus of fertilised eggs of FVB3 mice in cooperation with Macrogen (Seoul, Korea). Tail DNA was isolated and *SELENOW*-overexpressing transgenic FVB3 mice were identified by detecting the 859-bp fragments by PCR using the following primers: sense, 5′-TTCCAGTTCTGGGGAAGTCC-3′ and antisense, 5′-TCTAGGGCT-CACTGGCACTG-3′. Ectopic expression of *SELENOW* during differentiation of transgenic mouse osteoclast precursors into osteoclasts was confirmed by immunoblotting using an antibody against SELENOW. The animal protocol and experimental procedures were approved by the Institutional Animal Care and Use Committee of Yeungnam University College of Medicine and were in compliance with the Guide for the Care and Use of Laboratory Animals.

**µCT, and histological and histomorphometric analyses of bone.** After mice were sacrificed, the proximal tibia was scanned with a high-resolution 1076 µCT system (Skyscan, Aartselaar, Belgium) and bone indices including trabecular bone volume per tissue volume, trabecular bone number, trabecular thickness, trabecular separation, and bone mineral density were determined. For histological analyses, 5-µm-thick sagittal sections prepared on a microtome were stained with haematoxylin and eosin to detect osteoblasts or TRAP to visualise osteoclasts. In addition, the bone volume per total volume and eroded bone surface were measured from the section using TRAP staining. For bone histomorphometric analyses, mice were intraperitoneally injected with calcein (15 mg/kg) 10 and 3 days before killing. The diaphyseal cortical bone of the tibia was cut into 20-µm-thick sections using a grinder, and the calcein double-labelled bone surface was photographed to determine bone formation rate.

**Calvarial bone histomorphometric and bone resorption marker analyses.** Three-dimensional images of calvarial bone were constructed based on the µCT scan and bone indices were estimated. For analysis of calvarial bone histology, whole and cross-sectioned calvaria were stained with TRAP to visualise the extent of osteoclastogenic activity and to assess the number of mature osteoclasts, respectively. Calvarial bone sections were stained with haematoxylin and eosin and the calvarial marrow cavity was measured using Image-Pro Plus v.6.0 software (Media Cybernetics) to estimate the size of the osteolytic lesion. For biochemical analysis of osteoporosis, urinary DPD, a known bone resorption marker, was measured using a commercial immunoassay kit (MicroVeu DPD Enzyme Immunoassay; Quidel, San Diego, CA, USA) according to the manufacturer's instructions.

**Northern blotting, RT-PCR, quantitative real-time (q)PCR, and immunoblotting.** Total RNA was isolated from cells or tissues using TRIzol reagent (Invitrogen). *SELENOW* mRNA level was evaluated by northern blotting using a [$^{32}$P] dCTP-labelled DNA probe against *SELENOW* (5′-TCAAAGAACCCGGTGACC TG-3′) according to the conventional capillary method[71]. The 18S and 28S rRNA bands on the denaturing agarose gel stained with acridine orange were photographed and served as a loading control. To assess the mRNA levels of various genes, total RNA was reverse transcribed into cDNA using random hexamers and an MMLV-RT kit (Invitrogen) and target genes were amplified by RT-PCR using the primers listed in Supplementary Table 1. For qPCR analysis, total RNA was reverse transcribed into cDNA with the Superscript First-Strand Synthesis System (Invitrogen) and the reaction was carried out on a 7500 Detection System (Applied Biosystems, Foster City, CA, USA) using the Real-time TaqMan PCR assay kit that included primer sets for *NFATc1* (Mm00479445_m1), *Acp5* (Mm00475698_m1), and *OSCAR* (Mm00558665_m1) (Thermo Fisher Scientific). The mRNA level was normalised to that of *glyceraldehyde 3-phosphate dehydrogenase* and quantified with the comparative threshold cycle method. For immunoblotting, cells were lysed with lysis buffer [20 mM Tris-HCl (pH 7.5), 150 mM NaCl, 1% Nonidet (N)P-40, 0.5% sodium deoxycholate, 1 mM EDTA, and 0.1% sodium dodecyl sulphate (SDS)] containing a protease and phosphatase inhibitor cocktail (Roche, Indianapolis, IN, USA). The cell lysates were centrifuged, and protein concentration in the supernatant was measured with a detergent-compatible protein assay (Bio-Rad, Hercules, CA, USA). Cytosolic and nuclear proteins were fractionated as described[70]. Proteins were resolved by SDS-polyacrylamide gel electrophoresis (PAGE) on a 10% polyacrylamide gel and transferred to a nitrocellulose membrane that was probed with primary antibodies. Antibodies against SELENOW (Novus Biologicals, NBP1-49599; 1:1000), β-actin (Santa Cruz Biotechnology, sc-47778; 1:1000), NFATc1 (Santa Cruz Biotechnology, sc-47778; 1:1000), p65 (Santa Cruz Biotechnology, sc-372; 1:1000), 14-3-3 gamma (Santa Cruz Biotechnology, sc-398423; 1:1000), c-Jun (Santa Cruz Biotechnology, sc-1694; 1:1000), c-Fos (Cell Signaling, #4384; 1:1000), 6x-His Tag (Thermo Scientific, MA1-21315; 1:1000), c-Src (Santa Cruz Biotechnology, sc-8056; 1:1000), p-ERK (Cell Signaling, #9101; 1:1000), ERK (Cell Signaling, #9102; 1:1000), p-JNK (Cell Signaling, #9251; 1:1000),

JNK (Cell Signaling, #9252; 1:1000), p-p38 (Cell Signaling, #9211; 1:1000), p38 (Cell Signaling, #9212; 1:1000), and GAPDH (Abcam, ab8245; 1:1000) were purchased from the indicated suppliers. Whole-cell extracts to detect SELENOW were separated by SDS-PAGE on a 4–12% gradient gel (Invitrogen).

**Immunoblot analysis of SELENOW in various mouse tissues.** Mouse tissues including the heart, liver, lung, kidney, fat, pancreas, spleen, stomach, intestines, skin, skeletal muscle, brain, testis, lymph node, and long bone were washed three times with PBS and cut into small pieces in lysis buffer. After sonication with 30-s pulses on ice, tissue homogenates were centrifuged at $10,000 \times g$ for 10 min at 4 °C. The protein concentration of the supernatant was determined using a detergent-compatible protein assay. Proteins were fractionated by SDS-PAGE on a 4–12% gradient gel and immunoblotting was performed using an anti-SELENOW antibody.

**Luciferase reporter assay.** Murine monocytic RAW264.7 cells ($1 \times 10^5$ cells/well seeded in a 24-well plate) were cultured in α-MEM for 12 h. Luciferase reporter plasmids (NF-κB-, NFATc1-, and AP-1-dependent luciferase reporter, pcDNA3.1-His-tagged SELENOW (SeCys-13) or pcDNA3-β-gal) were transfected into the cells using Lipofectamine 2000. After stimulating the cells with RANKL for 24 h, luciferase activity was assessed with the dual-luciferase reporter assay system (Promega, Madison, WI, USA) and a luminometer (Turner Designs Hydrocarbon Instruments, Fresno, CA, USA), and normalised to β-galactosidase activity.

**RNA sequencing.** Osteoclast precursors obtained from SELENOW$^{-/-}$ and SELENOW-overexpressing transgenic mice and corresponding wild-type mice (C57BL6 and FVB3, respectively) were differentiated into osteoclasts in the presence of M-CSF and RANKL for 3 days. Total RNA was extracted using TRIzol reagent (Invitrogen) according to the manufacturer's instructions. For quality control, RNA purity and integrity were verified by measured the optical density ratio at 260/280 nm on a 2100 Bioanalyzer (Agilent Technologies, Santa Clara, CA, USA). Strand-specific library perpetration was carried out using an Illumina Truseq stranded mRNA library prep kit (Illumina, San Diego, CA, USA). Briefly, total RNA (1 μg) was reverse transcribed to cDNA using a random hexamer primer. Second-strand cDNA was synthesised, in vitro transcribed, and labelled by incorporation of dUTP. Validation of the library preparations was performed by capillary electrophoresis (2100 Bioanalyzer; Agilent). Libraries were quantified using the SYBR Green PCR Master Mix (Applied Biosystems). After adjusting the concentration to 4 nM, the libraries were pooled for multiplex sequencing. Pooled libraries were denatured and diluted to 15 pM and then clonally clustered onto the sequencing flow cell using the Illumina cBOT automated cluster generation system (Illumina). The clustered flow cell was sequenced with 2 × 100 bp reads on the HISEQ 2500 system (Illumina) according to the manufacturer's protocol. For bioinformatic analyses of RNA sequences, high-quality reads were aligned to the reference mm10 genome (http://support.illumina.com/sequencing/sequencing_software/igenome. html) using TopHat-2 v.2.0.13 (http://ccb.jhu.edu/software/tophat). The aligned reads were analysed with Cuffdiff v.2.2.0 (http://cole-trapnell-lab.github.io/cufflinks/cuffdiff/) to detect genes that were differentially expressed between cells expressing or deficient in SELENOW and control cells. Differential expression was estimated by selecting transcripts that showed significant changes ($P < 0.05$). For ontology analysis, differentially expressed genes were selected and processed using Gene Ontology (GO) consortium (http://geneontology.org) or DAVID (http://david.abcc.ncifcrf.gov) to obtain a comprehensive set of GO terms for biological processes. Based on biological processes in the Gene Ontology (GO) consortium, cell differentiation-related upregulated genes with a >2-fold increase and P-value < 0.05 were analysed before (day 0) and after (day 3) differentiation of wild-type (WT) osteoclast precursors into osteoclasts. The fold-induction of each cell differentiation-related gene (129 and 123 genes in osteoclasts from C57BL6 and FVB3 WT mice, respectively) was set to 1, and the relative fold change in the levels of genes in SELENOW$^{-/-}$ and SELENOW-overexpressing osteoclasts was determined

**Pull-down, IP, and ChIP assays.** For the pull-down assay, HEK 293T cells were transfected with pcDNA3.1 harbouring a His-tagged SELENOW (SeCys-13) and His-tagged SELENOW mutants in which SeCys-13 was replaced with cysteine or serine, using Lipofectamine 2000 and cell lysates were prepared with radio-immunoprecipitation assay (RIPA) buffer [50 mM Tris-HCl (pH 7.4), 1% NP-40, 150 mM NaCl, 0.25% sodium deoxycholate, 2 mM phenylmethylsulphonyl fluoride] containing complete protease inhibitor cocktail (Roche) 48 h after transfection. The lysate was centrifuged, and the supernatant was incubated overnight at 4 °C with an anti-His-tag antibody. After further incubation with protein G agarose beads for 3 h at 4 °C, immune complexes were washed three times with RIPA buffer and subjected to immunoblotting using indicated antibodies. For immunoprecipitation analysis, osteoclast precursors infected with SELENOW-overexpressing retrovirus were cultured with M-CSF and RANKL for 2 days, yielding pre-osteoclasts. Nuclear extracts from pre-osteoclasts were immunoprecipitated with antibodies against NFATc1, p65, 14-3-3γ, or SELENOW (Novus Biologicals). The ChIP assay was performed using the EZ-ChIP kit (Millipore, Billerica, MA, USA) according to the manufacturer's protocols. After the IP of chromatin with an antibody against SELENOW or control IgG, PCR was performed using DNA and promoter-specific primers containing the NF-κB- or NFATc1-binding sites listed in Supplementary Table S1.

**F-actin staining and caspase activity assay.** For detection of the actin ring, mature osteoclasts were transduced with SELENOW-overexpressing retrovirus, fixed with 3.7% formaldehyde, permeabilized with 0.1% Triton X-100 in PBS, and stained with fluorescein isothiocyanate-conjugated phalloidin (Sigma-Aldrich). Fluorescence images were acquired with a BX51 microscope (Olympus, Tokyo, Japan). TRAP + MNCs with a full actin ring were counted to assess osteoclast survival. To measure caspase activity in mature osteoclasts, osteoclast precursors were differentiated into osteoclasts in the presence of M-CSF and RANKL for 3 days. Cells were washed twice with PBS and treated with 0.1% trypsin-EDTA for 5 min to remove the monocytes. After washing with PBS, osteoclasts were incubated with enzyme-free cell dissociation solution (Millipore) for 30 min. Purified osteoclasts were re-plated at $1 \times 10^6$ cells on 10-cm culture dishes and then incubated for 24 or 36 h with M-CSF and RANKL. Caspase activity was assayed with a fluorometric kit (R&D Systems, Minneapolis, MN, USA). Briefly, the cells were washed with ice-cold PBS and lysed in the cell lysis buffer provided with the kit. The caspase-3 [DEVE-p-nitroaniline (pNA)] or caspase-9 (LEHD-pNA) substrate was added to the lysates in a 96-well plate followed by incubation for 1 h. The release of pNA was measured at 405 nm on a microplate reader.

**Measurement of thiol content.** For cellular thiol quantification, osteoclasts were washed with ice-cold PBS, followed by homogenisation and centrifugation. The resultant cytosolic fraction was incubated for 30 min at room temperature with 250 μM 5,5′-dithiobis-(2-nitrobenzoic acid) (Sigma-Aldrich) as previously described[72]. Total thiol content was measured by measuring the absorbance at 415 nm on a microplate reader (Model 680; Bio-Rad) using glutathione as the calibration standard. Data are represented as nanomoles of thiol per milligram of protein.

**Statistical analysis.** Data are presented as mean ± standard deviation (SD) of three independent experiments and were analysed with Prism 6 software (GraphPad Inc., La Jolla, CA, USA). Data were evaluated for normality and equal variance. Comparisons between two and multiple groups were performed with the Student's two-tailed $t$-test and one-way analysis of variance (ANOVA) with a post-hoc Tukey test, respectively. Statistical analysis was performed using SPSS 21.0 software. Differences were considered statistically significant at $P < 0.05$.

**Reporting summary.** Further information on research design is available in the Nature Research Reporting Summary linked to this article.

## Data availability

All data supporting the findings are presented within this paper and its Supplementary Information. Additional data that support the findings of this study are available from the corresponding author upon reasonable request. Source data are provided with this paper.

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

## Acknowledgements

We thank I.Y. Kim (Korea University) for pcDNA3.1 vector harbouring His-tagged SELENOW. This work was supported by grants from the National Research Foundation of Korea (Nos. 2016R1A2B2012108 and 2015R1A5A2009124).

## Author contributions

H.K. and D.J. initiated the study, designed experiments and analysed data. H.K. and K.L. performed and participated in most of the experiments. J.M.K. and J.-R.K. performed RNA-sequencing experiments and analysed the data. H.-W.L. designed and generated *SELENOW*$^{-/-}$ mice. M.Y.K performed the immunoprecipitation assay. Y.W.C. analysed the level of SELENOW in mouse tissues and helped with GeneChip arrays. H.-I.S., S.Y.L., and Y.C. performed histological analyses of bone and analysed bone parameters. T.K. analysed μCT. E.-S.P. and J.R. prepared osteoclast precursors from *TRAF6*$^{-/-}$ mice and performed in vitro experiments. S.H.L. performed and analysed qPCR experiments. N.K. helped with *SELENOW* transgenic mice construction and provided pBluescript vector containing TRAP promoter. H.K., K.L., and D.J. wrote the manuscript. All authors reviewed and edited the manuscript.

## Competing interests

The authors declare no competing interests.
