## [Peer Review File · Nature Communications]

Reviewers' Comments:

Reviewer #1:

Remarks to the Author:

In this manuscript, the authors investigated a role of SELENOW in bone homeostasis. The authors found that SELENOW-deficient mice displayed osteopetrosis whereas SELENOW-overexpressing mice exhibited osteoporosis, and concluded that RANKL-induced suppression of SELENOW in osteoclasts is crucial for the proper formation of osteoclasts in vivo. In vitro experiments suggested that SELENOW promotes osteoclast differentiation, fusion, function and survival. This manuscript is potentially interesting because this is the first report showing that the factor whose expression is suppressed during osteoclastogenesis positively regulates osteoclast differentiation. However, the present study has several critical concerns. First, the authors did not use osteoclast-specific SELENOW-deficient mice. Second, the function of SELENOW in osteoclasts and the underlying mechanisms are unclear. Third, the molecular target of SELENOW in osteoclasts is unclear.

Taken together, this manuscript is not suitable for publication in Nature Communications in its current form.

Major concerns

1. Bone metabolism is regulated by various systems such as nervous system, immune system and endocrine system. Given that SELENOW is ubiquitously expressed, the importance of osteoclast-intrinsic function of SELENOW in bone homeostasis cannot be proven without conditional KO mice. The authors should show the phenotypes of osteoclast-specific SELENOW-deficient mice.
2. Since this is the first report showing the phenotypes of SELENOW-deficient and SELENOW-overexpressing mice, the authors should provide more detailed phenotypes that are typically analyzed in the field of bone biology (e.g. body size, body weight, tooth eruption, serum calcium and phosphorus levels, whole-body X-ray picture, bone morphometric analysis and CT analysis of spine)
3. Based on in vitro experiments, the authors proposed that SELENOW promotes osteoclast differentiation, fusion, function and survival. However, in vivo, the authors just showed the number of osteoclasts and concluded that SELENOW is critical for proper osteoclast differentiation without monitoring other aspects of osteoclasts. Which function(s) of SELENOW is crucial for bone remodeling in vivo? To identify the function of SELENOW in osteoclasts in vivo, the authors should investigate multiple osteoclastic parameters (e.g. osteoclast size, eroded bone surface) in osteoclast-specific SELENOW-deficient mice as well as SELENOW-overexpressing mice.
4. The function of SELENOW in osteoclasts and the underlying mechanisms are unclear. The authors demonstrated that SELENOW promotes osteoclast differentiation by directly interacting with NFATc1 and NF- κ B, promoting the nuclear translocation of these key transcriptional factors. The authors also proposed that SELENOW promotes osteoclast fusion and function, and suppresses the apoptosis of mature osteoclasts by modulating the redox status of osteoclasts. The authors should clarify the molecular target of SELENOW in osteoclasts using various SELENOW mutants that lack NFATc1-interacting domain, NF- κ B-interacting domain as well as an antioxidant domain. The authors should also clarify how the direct interaction of SELENOW with NFATc1 and NF- κ B promotes their nuclear translocation.
5. NFATc1 and NF- κ B play crucial roles not only in osteoclasts but also in various cell types. If SELENOW is crucial for NFATc1 and NF- κ B pathway, it is expected that SELENOW-deficient mice display multiple phenotypes. The authors should provide more comprehensive information about the phenotypes of SELENOW-deficient mice.

6. The authors should explain why BFR is suppressed in SELENOW-deficient and SELENOW-overexpressing mice.

7. The authors should check the expression of OPG and RANKL in SELENOW-deficient and SELENOW-overexpressing osteoblasts. The authors should also check the osteoclastogenesis in a co-culture system using SELENOW-deficient and SELENOW-overexpressing osteoblasts.

Reviewer #2:

Remarks to the Author:

Based on the identification of selenoprotein W (Selenow) as a gene that is repressed by the pro-osteoclastogenic cytokine Rankl, the authors have performed in vitro and in vivo studies to demonstrate a critical role of Selenow in bone resorption. They found that Selenow enhances osteoclastogenesis, which was also confirmed in Selenow-deficient and Selenow-over-expressing mice. The major conclusion of the authors is therefore, that Rankl-dependent Selenow repression is required the limit osteoclastogenesis and excessive bone resorption. Overall this is a strong manuscript describing novel findings with some mechanistic insights. There are however several issues that need to be addressed, before the manuscript is acceptable for publication.

Specific comments:

- 1) As stated in the Introduction, there are various reports showing that selenium insufficiency negatively affects the skeleton, whereas Selenow-deficient mice display increased bone mass. Although this apparent inconsistency is probably explainable, the authors should discuss their findings in relation to the studies cited as References 22-29. Moreover, since selenium-deficiency in rats was found to affect serum parameters of calcium homeostasis (ref. 24), it might be worthwhile to determine serum PTH and 1,25-D3 in Selenow-deficient mice, especially since Selenow is ubiquitously expressed (Suppl. Fig. 1).
- 2) The absence of Selenow expression in Selenow-deficient mice and osteoclasts has to be shown.
- 3) One of the key findings of the study is the differential impact of Selenow-deficiency and over-expression on osteoclast number per bone surface. However, when comparing the wildtype values in Fig. 2d and Fig. 3f, there is a moderate difference, which is not neglectable, since the transgenic mice have the same osteoclast number as the wildtype controls for the Selenow-deficient mice. It is therefore extremely important to provide information about age and gender of the analyzed mice. Moreover, the authors have to state clearly that they have used littermate controls for analysis.
- 4) Although this is not the primary focus of the study, it is somehow confusing that the bone formation rate is reduced in both, Selenow-deficient and Selenow-over-expressing mice. Can the authors rule out a cell-autonomous of Selenow-deficient osteoblasts? Is the expression of the transgene truly osteoclast-specific? In any case, the authors should discuss how this aspect of the phenotypes might be explained.
- 5) Since the molecular influence of Selenow on the cellular redox status and osteoclast apoptosis appears plausible, the authors should analyze in vivo, whether there are significant differences in osteoclast apoptosis on bone sections from Selenow-deficient and Selenow-over-expressing mice.
- 6) The authors state that it would be informative to perform additional studies in order to develop "anti-osteoporotic agents using Selenow". Given the fact, that Selenow is primarily localized in the cytoplasm in a large number of different cell types or organs, this seems to be rather difficult. However, if the authors have specific ideas how their knowledge could translate into therapy, they should state this more precisely in the Discussion section.

Reviewer #3:

Remarks to the Author:

This is an interesting manuscript detailing a potential role for RANKL induced downregulation of SELENOW promotes maintenance of proper osteoclast differentiation, while not affecting osteoblasts. There are some issues with controls, missed opportunities for in vivo insights, and technical approaches. Comments are below.

General issues:

1. For the gene chip arrays and RT-PCR, what was the up/down regulation compared to (csf only, no Rankl?)
2. Western blot for selenow in figure 1a along with RNA levels, but why no western for the rest of the osteoclast specific markers?
3. There are some peculiarities of the paper such as the use of RT-PCR followed by agarose gels instead of the current standard method of real-time PCR to quantify.
4. The effects of SELENOW KO in vivo were surprisingly small for many of the figures (e.g. Fig 2d and e show about 30% less; partsof Fig 3 show small effects).
5. I don't really see how this study shows molecular mechanism, other than binding to certain proteins and somehow SELENOW reduces cytosolic redox status. Is the role of SELEONW to prevent oxidation of specific binding partners? The way experiments are conducted all of the effects could be either direct or indirect.
6. For the in vivo portion there were issues, and properly controlled experiments would allow better interpretation. For instance, in humans there are large discrepancies in rates of osteoporosis in males and females, I would have liked to have seen both sexes separately used (although they did specify they used male mice). Furthermore, bone density can be highly positivity correlated with body mass, I would like to have seen controls for body weight. And also they should have compared litter mates as is required for top tier journals like Nature Communications.
7. Fig. 5 g,h i all show that when selenow is overexpressed, there is an observed increase in reduction capacity in osteoclasts and increased survival (well they don't do an apoptosis assay but instead just count number of cells compared to control). The total thiol experiments in Fig. 5 involve unusually high levels of SELENOW and likely to not reflect physiologically relevant roles for the endogenous protein. Overexpression of any selenoprotein is known to do this, and the investigators need to come up with a better way to determine molecular mechanisms.
8. There seems to be missed opportunities on studying the mice involved in this manuscript for bone disease. The tissues were mainly studied ex vivo, but are there stress tests or other in vivo tests that could provide interesting insights into the outcome of loss-of-function and gain-of-function of SELENOW in mice?

Specific comments

1. Line 47, 56, 61, 147 : replace semi-colon with a period.
2. Line 88 : Need one or two more sentences describing the GeneChip arrays. What type of cells? How were they stimulated for how long? What was compared (e.g. unstimulated s stimulated)?
3. The data in Fig. 1 b are not conclusive and should be performed using real-time PCR. Also, the lack of an effect of IFNg on SELENOW levels may be due to mistiming of adding the IFNg or other aspects of the experimental approach. Were different pre-incubation periods with IFNg attempted before the RANKL was added. These types of negative data are not conclusive as presented without controls that rule out procedural artifacts.
4. Line 137; data not shown should be submitted as data in this paper. Characterization of these mice is crucial for the following experiments and the reader needs to be convinced the tissues are affected by the genetic manipulations employed. Need to show a tissue blot with various tissues analyzed for SELENOW.

5. Fig. 2b, real-time PCR for mRNA relative to which housekeeping mRNA?

6. LINE 177 change signalling to signaling

7. Fig 4a shows nuclear vs. cytosolic NFkB, but need to do EMSA to show it is functional. The reporter assays in 4b support this idea, but EMSA is the gold standard for NFkB translocation and binding to DNA.

8. Line 188 artefact change to artifact

9. Line 251 "redox status was reduced..." is not clear. Do you mean shifted toward an oxidized state?

10. Line 254 "SELENOW" change to SELENOW

Reviewer #1 (Remarks to the Author):

In this manuscript, the authors investigated a role of SELENOW in bone homeostasis. The authors found that SELENOW-deficient mice displayed osteopetrosis whereas SELENOW-overexpressing mice exhibited osteoporosis, and concluded that RANKL-induced suppression of SELENOW in osteoclasts is crucial for the proper formation of osteoclasts in vivo. In vitro experiments suggested that SELENOW promotes osteoclast differentiation, fusion, function and survival. This manuscript is potentially interesting because this is the first report showing that the factor whose expression is suppressed during osteoclastogenesis positively regulates osteoclast differentiation. However, the present study has several critical concerns. First, the authors did not use osteoclast-specific SELENOW-deficient mice. Second, the function of SELENOW in osteoclasts and the underlying mechanisms are unclear. Third, the molecular target of SELENOW in osteoclasts is unclear.

Taken together, this manuscript is not suitable for publication in Nature Communications in its current form.

Major concerns:

1. Bone metabolism is regulated by various systems such as nervous system, immune system and endocrine system. Given that SELENOW is ubiquitously expressed, the importance of osteoclast-intrinsic function of SELENOW in bone homeostasis cannot be proven without conditional KO mice. The authors should show the phenotypes of osteoclast-specific SELENOW-deficient mice.

Response: Thank you for your suggestion. We analysed the phenotypes of osteoclast-specific SELENOW-deficient mice. To produce osteoclast-specific SELENOW-deficient mice, animals carrying the floxed conditional SELENOW^{tm1c} allele in the C57BL/6 background were purchased from the European Mouse Mutant Archive (EMMA, ID EM:09166, Wellcome Trust Sanger Institute, Prague, Czech Republic). LysM-Cre mice (Stock Number: 004781) in the C57BL/6 background were obtained from Jackson Laboratory (Bar Harbor, USA). The conditional SELENOW^{tm1c/tm1c} mice were bred with LysM-Cre mice in order to generate mice deficient for SELENOW in monocytes/macrophages (LysM-Cre;SELENOW^{tm1c/tm1c}). SELENOW^{tm1c/tm1c} mice were used as wild-type littermate control.

Microcomputed tomography (μ CT) analysis of trabecular bone in the proximal tibia demonstrates that osteoclast-specific SELENOW-deficient mice (LysM-Cre;SELENOW^{tm1c/tm1c}) showed an increase in bone mass caused by an increase in trabecular bone mineral density, bone volume, trabecular number and trabecular thickness, and a decrease in trabecular bone separation when compared with wild-type littermate mice (SELENOW^{tm1c/tm1c}), as shown in Fig. 2d-g (newly assigned in the revision). Histomorphometric analysis of LysM-Cre;SELENOW^{tm1c/tm1c} also exhibited decreased TRAP-positive osteoclasts, osteoclast size, and eroded bone surface on the surface of trabecular bone when compared with littermate control mice (SELENOW^{tm1c/tm1c}). Taken together, our new results demonstrate that mice with SELENOW deficiency in osteoclasts showed an increase in bone mass, which is consistent with the osteopetrotic phenotype of whole-body SELENOW knockout mice (SELENOW^{-/-} mice).

2. Since this is the first report showing the phenotypes of SELENOW-deficient and SELENOW-overexpressing mice, the authors should provide more detailed phenotypes that are typically analyzed in the field of bone biology (e.g. body size, body weight, tooth eruption, serum calcium and phosphorus levels, whole-body X-ray picture, bone morphometric analysis and CT analysis of spine).

Response: Thank you for your comment. We analysed more detailed phenotypes of SELENOW-deficient and SELENOW-overexpressing mice. No significant differences were observed in whole-body X-ray images of the skeleton, tooth eruption, as well as calcium and phosphate serum levels between the SELENOW-deficient (SELENOW^{-/-}) mice and wild type littermates (Supplementary Fig. 5). SELENOW-overexpressing mice had also no significant changes in whole-body X-ray images, tooth eruption, and serum levels of calcium and phosphate when compared with their wild type littermates (Supplementary Fig. 8).

Body weight of SELENOW^{-/-} mice was higher than that of wild type littermates (27.77 ± 1.06 g versus 24.65 ± 1.74 g, $P < 0.01$; Supplementary Fig. 5a), and SELENOW-overexpressing mice exhibited a reduced

body weight compared with their wild type littermates (28.01 ± 1.63 g versus 30.91 ± 2.23 g, $P < 0.01$; Supplementary Fig. 8a).

In addition, we conducted μ CT and bone morphometric analysis of trabecular bone in the lumbar vertebra (L4) of SELENOW^{-/-} and SELENOW-overexpressing mice (Supplementary Fig. 6 and 9). Trabecular bone volume per tissue volume (BV/TV), trabecular number (Tb.N), and bone mineral density (BMD) of lumbar spine were higher in SELENOW^{-/-} mice in comparison to their corresponding wild type littermates. In contrast, trabecular bone volume per tissue volume, trabecular number, and bone mineral density of lumbar spine were lower in SELENOW-overexpressing mice than in corresponding wild type littermates. These results obtained from lumbar spine were consistent with those from proximal tibia.

3. Based on *in vitro* experiments, the authors proposed that SELENOW promotes osteoclast differentiation, fusion, function and survival. However, *in vivo*, the authors just showed the number of osteoclasts and concluded that SELENOW is critical for proper osteoclast differentiation without monitoring other aspects of osteoclasts. Which function(s) of SELENOW is crucial for bone remodeling *in vivo*? To identify the function of SELENOW in osteoclasts *in vivo*, the authors should investigate multiple osteoclastic parameters (e.g. osteoclast size, eroded bone surface) in osteoclast-specific SELENOW-deficient mice as well as SELENOW-overexpressing mice.

Response: To identify SELENOW function in osteoclasts *in vivo*, we analysed osteoclastic parameters such as osteoclast size and eroded bone surface in osteoclast-specific SELENOW-deficient mice as well as SELENOW-overexpressing mice. Osteoclast size and eroded bone surface of both whole-body SELENOW-deficient mice (SELENOW^{-/-} mice) and osteoclast-specific SELENOW-deficient mice (LysM-Cre;SELENOW^{tm1c/tm1c}) were lower than those of wild type littermates (Fig. 2c, f). On the contrary, SELENOW-overexpressing mice exhibited an increased osteoclast size and eroded bone surface compared with their wild type littermates (Fig. 3d). These results showed that SELENOW plays an important role in the differentiation and function of osteoclasts *in vivo*.

4. The function of SELENOW in osteoclasts and the underlying mechanisms are unclear. The authors demonstrated that SELENOW promotes osteoclast differentiation by directly interacting with NFATc1 and NF- κ B, promoting the nuclear translocation of these key transcriptional factors. The authors also proposed that SELENOW promotes osteoclast fusion and function and suppresses the apoptosis of mature osteoclasts by modulating the redox status of osteoclasts. The authors should clarify the molecular target of SELENOW in osteoclasts using various SELENOW mutants that lack NFATc1-interacting domain, NF- κ B-interacting domain as well as an antioxidant domain. The authors should also clarify how the direct interaction of SELENOW with NFATc1 and NF- κ B promotes their nuclear translocation.

Response: Thank you for your comment and assessment. Selenoproteins contain a selenocysteine (SeCys) amino acid residue translated from the in-frame UGA codon that is normally recognized as a stop codon (Nature, 2000, 407:463-465). The SeCys insertion sequence (SECIS) element in the 3' untranslated region (UTR) of eukaryotic selenoprotein-encoding mRNAs is an RNA element with approximately 60 nucleotides in length and contains a conserved stem-loop structure, which is essential for the decoding of UGA codon as a SeCys. Since the secondary structure of the SECIS element in SELENOW mRNA is required for successful translation of UGA as SeCys and Selenoprotein W is the smallest mammalian selenoprotein (89 amino acids), it is very difficult to construct a series of sequential SELENOW deletion mutants lacking NFATc1-interacting domain and NF- κ B-interacting domain as well as an antioxidant domain. Instead, we thought that a SeCys at codon 13 plays an important role in the interaction of SELENOW with NFATc1 and NF- κ B. To investigate the interaction of SELENOW with NFATc1 and NF- κ B, we used cells with SELENOW mutant in which a SeCys is substituted with a cysteine (SeCys13C) or a serine (SeCys13S). Both mutants, SELENOW (SeCys13C) and SELENOW (SeCys13S), were unable to interact with either NFATc1 or NF- κ B (Fig. 4d), indicating that a SeCys at codon 13 plays a critical role in the interaction between SELENOW and its targets including NFATc1 and NF- κ B. In addition, we previously showed that a SeCys at codon 13 is crucial for the antioxidant activity of SELENOW (FEBS Letters, 2002, 517:225-228). Together, a SeCys at codon 13 of SELENOW is thought to be critical for the interaction of SELENOW with NFATc1 and NF- κ B as well as for its antioxidant activities.

Based on the result which indicates that a SeCys-13 residue of SELENOW plays a part in the binding of SELENOW to NFATc1 or NF- κ B, as well as new additional experimental data at the revision stage and previously reported, we described a possible mechanistic role of SELENOW in osteoclast differentiation in the revised Discussion as follows:

“A specific interaction between SELENOW and 14-3-3 protein is known to involve the conserved redox motif CysXXSeCys present in a SELENOW exposed loop, and an exposed Cys residue in the C-terminal domain of 14-3-3 protein (Biochemistry, 2007, 46:6871-6882). The binding of SELENOW to 14-3-3 protein was significantly enhanced under oxidative stress conditions (Biochim Biophys Acta, 2016, 1863:10-18), indicating that the intracellular redox status regulates SELENOW-14-3-3 protein interactions, and modulates 14-3-3 protein functional activity. Here, we observed that the SELENOW SeCys-13 residue is critical for the interaction with its targets, NFATc1 and NF- κ B, and that SELENOW leads to an increase in cell reduction capacity. Also, we previously highlighted that this residue, SeCys-13, is crucial for SELENOW antioxidant activity (FEBS Letters, 2002, 517:225-228). Altogether, these findings suggest that SELENOW has the potential to enhance osteoclast differentiation via direct interaction with NFATc1 and NF- κ B and their subsequent activation, but also indirectly, as a reducing agent.”

Additionally, we have indicated “negative feedback mechanism of osteoclastogenic SELENOW in osteoclast differentiation” in the first section of the Discussion. We hope the reviewer will accept the rationale of our work upon assessing our entire manuscript as a whole.

5. NFATc1 and NF- κ B play crucial roles not only in osteoclasts but also in various cell types. If SELENOW is crucial for NFATc1 and NF- κ B pathway, it is expected that SELENOW-deficient mice display multiple phenotypes. The authors should provide more comprehensive information about the phenotypes of SELENOW-deficient mice.

Response: Thank you for your comment. We agree with your assessment. Bone-resorbing osteoclasts and bone-forming osteoblasts are the two major cells for bone remodelling. It has been well documented that NFATc1 and NF- κ B play a crucial role in both osteoclast and osteoblast differentiation (Nat Med, 2007, 13:791-801). The activation of NFATc1 and NF- κ B was required for sufficient osteoclast differentiation; however, NFATc1 and NF- κ B negatively regulate osteoblast differentiation.

During osteoclast differentiation, SELENOW was found to be downregulated via RANKL/RANK/TRAF6/p38 signalling. Conversely, we demonstrated that SELENOW expression was not altered during osteoblast differentiation, and that modulating SELENOW levels had no effect on osteoblast differentiation or the expression of osteoblast differentiation markers such as alkaline phosphatase and osteopontin. Taken together, we showed that SELENOW regulates NFATc1 and NF- κ B activities in a RANKL-dependent manner during osteoclast differentiation, but not during osteoblast differentiation, which suggests that the regulation of NFATc1 and NF- κ B by SELENOW may occur distinctively in different cell types.

In addition, we analysed more detailed phenotypes of SELENOW-deficient (SELENOW^{-/-}) mice. No significant differences were observed in whole-body X-ray analysis for the skeleton, tooth eruption, and serum levels of calcium and phosphate between SELENOW^{-/-} mice and wild type littermates (Supplementary Fig. 5b, c, e). To investigate neuromotor coordination and balance, mice were analysed by rotarod experiments. In the accelerating rotarod, no differences in latency to fall from the test apparatus were observed between SELENOW^{-/-} mice and wild type littermates (Supplementary Fig. 5d). We have also conducted the open field test to study the overall locomotor activities of SELENOW^{-/-} mice (Supplementary Fig. 5d). There were no significant differences in the total travel distances between SELENOW^{-/-} mice and wild type littermates. In conclusion, results on the phenotypes of SELENOW^{-/-} mice indicate the osteoclast-specific effect of SELENOW on the regulation of NFATc1 and NF- κ B.

6. The authors should explain why BFR is suppressed in SELENOW-deficient and SELENOW-overexpressing mice.

Response: Thank you for your suggestion. In our original manuscript, we showed that global deletion of SELENOW in mice resulted in a decrease in bone formation rate (BFR) and presented an osteopetrotic phenotype. In the revised version of this manuscript, we generated osteoclast-specific SELENOW-deficient

mice (*LysM-Cre;SELENOW^{tm1c/tm1c}*) to verify decreased BFR by osteoclast defect. *LysM-Cre;SELENOW^{tm1c/tm1c}* showed a decreased BFR compared with littermate control mice (*SELENOW^{tm1c/tm1c}*), as shown in Fig. 2g. Osteopetrosis, due to lack of osteoclasts, has been reported to be characterised by reduced bone formation (Calcif Tissue Int, 2014, 95:83-93). Thus, our results clearly demonstrate the effect of SELENOW deletion on BFR, showing that SELENOW deficiency results in the suppression of osteoclastogenesis and the reduction of BFR.

Also, in our original manuscript, we generated transgenic mice in which SELENOW gene expression was controlled by the promoter of tartrate-resistant acid phosphatase (TRAP). Although TRAP expression in bones was attributed to osteoclasts and their precursor cells, multiple tissues such as kidney, liver, lung, pancreas, spleen, bone, and intestine have been known to highly express TRAP (J Histochem Cytochem, 2000, 48:219-228; J Bone Miner Res, 2000, 15:103-110). Since it may be difficult to study the bone-specific effect of SELENOW overexpression on BFR, we decided to remove BFR data in SELENOW-overexpressing mice to avoid confusion.

7. The authors should check the expression of OPG and RANKL in SELENOW-deficient and SELENOW-overexpressing osteoblasts. The authors should also check the osteoclastogenesis in a co-culture system using SELENOW-deficient and SELENOW-overexpressing osteoblasts.

Response: Thank you for your comment. We analysed the expression of OPG and RANKL in osteoblasts derived from SELENOW-deficient and SELENOW-overexpressing mice. Results obtained by real-time PCR showed no significant differences in the expression levels of OPG and RANKL between osteoblasts from SELENOW-deficient mice and wild type littermates as well as between osteoblasts from SELENOW-overexpressing mice and wild type littermates (Supplementary Fig. 10a, b). In addition, we analysed osteoclast differentiation in a co-culture system using osteoblasts prepared from SELENOW-deficient and SELENOW-overexpressing mice. Our studies on co-culture system of osteoclast precursors and osteoblasts revealed that there were no changes in the number of TRAP-positive multinucleated cells having more than 10 nuclei between the co-culture of osteoblasts from SELENOW-deficient mice and osteoclast precursors from wild type littermates and the co-culture of osteoblasts from wild type littermates and osteoclast precursors from wild type littermates. Also, no changes were observed in the number of TRAP-positive multinucleated cells between the co-culture of osteoblasts from SELENOW-overexpressing mice and osteoclast precursors from wild type littermates and the co-culture of osteoblasts from wild type littermates and osteoclast precursors from wild type littermates (Supplementary Fig. 10c, d). In conclusion, regardless of expression levels of SELENOW in osteoblasts, osteoclast precursors derived from SELENOW-deficient and SELENOW-overexpressing mice displayed respective decreased and increased osteoclast differentiation. Furthermore, we demonstrated that osteoclast-specific SELENOW-deficient mice (*LysM-Cre;SELENOW^{tm1c/tm1c}*) showed an increased bone mass resulting from elevation in trabecular bone mineral density, bone volume, trabecular number and trabecular thickness, and decreased trabecular bone separation when compared with littermate control mice (*SELENOW^{tm1c/tm1c}*), as presented in Fig. 2. Taken together, we suggest that the regulatory effect of SELENOW on bone metabolism is caused by osteoclast-intrinsic function, but not by osteoblasts.

Reviewer #2 (Remarks to the Author):

Based on the identification of selenoprotein W (Selenow) as a gene that is repressed by the pro-osteoclastogenic cytokine Rankl, the authors have performed in vitro and in vivo studies to demonstrate a critical role of Selenow in bone resorption. They found that Selenow enhances osteoclastogenesis, which was also confirmed in Selenow-deficient and Selenow-over-expressing mice. The major conclusion of the authors is therefore, that Rankl-dependent Selenow repression is required to limit osteoclastogenesis and excessive bone resorption. Overall this is a strong manuscript describing novel findings with some mechanistic insights. There are however several issues that need to be addressed, before the manuscript is acceptable for publication.

Specific comments:

1-1) As stated in the Introduction, there are various reports showing that selenium insufficiency negatively affects the skeleton, whereas Selenow-deficient mice display increased bone mass. Although this apparent inconsistency is probably explainable, the authors should discuss their findings in relation to the studies cited as References 22-29.

Response: Thank you for your evaluation and comments. Selenium was known to be an essential trace element in mammals and its physiological functions are primarily mediated through selenium-containing proteins, selenoproteins (Biochemistry, 2007, 46:6871-6882; Biochim Biophys Acta, 2014, 1840:3246-3256). There are 25 and 24 known selenoproteins in humans and rodents, respectively. Most of selenoproteins, including glutathione peroxidases and thioredoxin reductases, play an important role in maintaining cellular antioxidant homeostasis, as stated in the Introduction section.

Oxidative stress mediated by reactive oxygen species (ROS) has been shown to be deleterious to normal bone physiology. Excessive ROS production was found to not only induce osteoclast differentiation and inhibit osteoblast differentiation (Endocr Rev, 2010, 31:266-300; Trends Mol Med, 2009, 15:468-477), but also promote the expression of osteoclastogenic factor RANKL in osteoblasts (J Biol Chem, 2005, 280:17497-17506). A number of antioxidant selenoenzymes, including glutathione peroxidases and thioredoxin reductases, are known to be involved in both induction of osteoblast differentiation and inhibition of osteoclast differentiation by reducing intracellular ROS levels (Biochim Biophys Acta, 2014, 1840:3246-3256; Biol Trace Elem Res, 2012, 150:441-450; Free Radic Biol Med, 2011, 51:1533-1542). Therefore, the deficiency of selenium and selenoproteins in bone could potentially have negative effects on bone metabolism and result in impaired skeletal development due to the failure to suppress oxidative stress in both osteoblasts and osteoclasts (J Nutr, 2012, 142:1526-1531; J Bone Miner Res, 2001, 16:1556-1563). In the present study, we showed that whole-body SELENOW-deficient and osteoclast-specific SELENOW-deficient mice exhibited osteopetrosis in which bone resorption was impaired due to osteoclast dysfunction, with no changes in osteoblast activity.

In conclusion, negative impacts on bone homeostasis by selenium insufficiency could be considered as a consequence of failure to reduce oxidative stress, resulting in inhibition of osteoblast differentiation and stimulation of osteoclast differentiation. Conversely, the osteopetrotic phenotype of SELENOW-deficient mice is resulted from the attenuated osteoclastic activity with no alteration in osteoblastic activity. As reviewer recommended, this point was described in the Discussion section.

1-2) Moreover, since selenium-deficiency in rats was found to affect serum parameters of calcium homeostasis (ref. 24), it might be worthwhile to determine serum PTH and 1,25-D3 in Selenow-deficient mice, especially since Selenow is ubiquitously expressed (Suppl. Fig. 1).

Response: We determined serum levels of parathyroid hormone (PTH) and 1,25-dihydroxyvitamin D3 using ELISA assay. SELENOW-deficient mice and SELENOW-overexpressing mice showed no significant differences in serum levels of PTH, 1,25-dihydroxyvitamin D3, calcium, and phosphate when compared with their wild type littermates (Supplementary Fig. 5e and 8e).

2) The absence of Selenow expression in Selenow-deficient mice and osteoclasts has to be shown.

Response: Thank you for your suggestion, we have included results showing the absence of SELENOW expression in SELENOW-deficient mice and osteoclasts (Supplementary Fig. 4a, b, c)

3) One of the key findings of the study is the differential impact of Selenow-deficiency and over-expression on osteoclast number per bone surface. However, when comparing the wildtype values in Fig. 2d and Fig. 3f, there is a moderate difference, which is not neglectable, since the transgenic mice have the same osteoclast number as the wildtype controls for the Selenow-deficient mice. It is therefore extremely important to provide information about age and gender of the analyzed mice. Moreover, the authors have to state clearly that they have used littermate controls for analysis.

Response: Thank you for your comment. We stated in Methods section of our original manuscript that two genetically modified mice, SELENOW^{-/-} C57BL6 mice and SELENOW-overexpressing transgenic FVB3 mice, were used to analyse the role of SELENOW in bone homeostasis. Two different wild type mice, C57BL6 and FVB3, were used to generate SELENOW-deficient and SELENOW-overexpressing mice, respectively. Bone parameters such as trabecular bone volume per tissue volume (BV/TV), trabecular number, trabecular thickness, and trabecular separation have been shown to be varied in mice strains (J Bone Miner Res, 2005, 20:1085-1092), which can explain the difference in the number of osteoclasts per bone surface between C57BL6 and FVB3 wild type mice. In addition, ten-week-old male mice of SELENOW-deficient mice, SELENOW-overexpressing mice, and their wild type littermates were analysed in our study. This is described in Methods section. We believe that this reflects a fundamental difference in the nature of two the mice strains.

4) Although this is not the primary focus of the study, it is somehow confusing that the bone formation rate is reduced in both, Selenow-deficient and Selenow-over-expressing mice. Can the authors rule out a cell-autonomous of Selenow-deficient osteoblasts? Is the expression of the transgene truly osteoclast-specific? In any case, the authors should discuss how this aspect of the phenotypes might be explained.

Response: Thank you for your comment. In our original manuscript, we showed that global deletion of SELENOW in mice resulted in a decrease in bone formation rate (BFR) and presented an osteopetrotic phenotype. In the revised version of this manuscript, we generated osteoclast-specific SELENOW-deficient mice (*LysM-Cre;SELENOW^{tm1c/tm1c}*) to verify decreased BFR by osteoclast defect. *LysM-Cre;SELENOW^{tm1c/tm1c}* showed a decreased BFR compared with littermate control mice (*SELENOW^{tm1c/tm1c}*), as shown in Fig. 2g. Thus, we are able to rule out a cell-autonomous effect of SELENOW-deficient osteoblasts on the regulation of bone metabolism and our data clearly demonstrate the effect of SELENOW deletion on BFR, showing that SELENOW deficiency results in the suppression of osteoclastogenesis and the reduction of BFR.

Conversely, we generated transgenic mice in which SELENOW gene expression was controlled by the promoter of tartrate-resistant acid phosphatase (TRAP). Although TRAP expression in bone is attributed to osteoclasts and their precursor cells, multiple tissues such as kidney, liver, lung, pancreas, spleen, bone, and intestine have been known to highly express TRAP (J Histochem Cytochem. 2000, 48:219-228; J Bone Miner Res, 2000, 15:103-110). Since it may be difficult to study the bone-specific effect of SELENOW overexpression on BFR, we decided to remove BFR data in SELENOW-overexpressing mice to avoid confusion.

5) Since the molecular influence of Selenow on the cellular redox status and osteoclast apoptosis appears plausible, the authors should analyze *in vivo*, whether there are significant differences in osteoclast apoptosis on bone sections from Selenow-deficient and Selenow-over-expressing mice.

Response: Thank you for your suggestion. We tried to analyse osteoclastic cell apoptosis on the surface of trabecular bone from SELENOW-deficient (SELENOW^{-/-}) and SELENOW-overexpressing (TG) mice. For the detection of apoptotic osteoclast cells, after the tibial trabecular bone from SELENOW^{-/-} and TG mice was decalcified for 2 weeks, 5- μ m-thick sagittal sections were prepared on a microtome and we performed *in vivo* terminal deoxynucleotidyl transferase-mediated deoxyuridine triphosphate nick end labeling (TUNEL) assay using the TACS 2TdT DAB In Situ Apoptosis Detection Kit (Trevigen Inc., MD, USA) and In Situ Cell Death Detection Kit, AP (Cat.No.11 684 809 910, Roche, Germany). The colour and fluorescence sections stained with TUNEL were visualized under respective light and fluorescence microscope (Fig. R1). Also, the sequential sections were stained with TRAP to detect osteoclasts (data not shown). Unfortunately, we could not discriminate the apoptotic osteoclasts from other numerous non-osteoclastic cells in overlapped images between TUNEL-positive apoptotic cells and TRAP-positive osteoclasts. Some reasons for detecting apoptotic

osteoclasts in bone sections include: i) osteoclasts are specialized multinucleated giant cells derived from monocyte fusion and have about 2 to 100 nuclei per cell (Endoc Rev, 1996, 17:308-332), and it is difficult to distinguish the apoptotic images of osteoclasts with dispersed multi-nuclei from those of non-osteoclastic cells with mono-nucleus. ii) As previously reported (Endoc Rev, 2000, 21:115-137), the terminally differentiated mature osteoclasts that resorb the bone have a short life span of approximately 2 weeks, whereas the life span of osteoblasts is approximately 3 months. A short retention time of osteoclasts *in vivo* will make it difficult to observe the apoptotic osteoclasts. iii) Decalcification process for a period of 2 weeks to produce bone sections may cause destruction of apoptotic factors and interfere with the detection of apoptotic osteoclast cells. Instead of apoptotic data of osteoclasts, we analysed osteoclast size and eroded bone surface, which reflect the existence of bone-resorbing osteoclasts and osteoclastic function, in trabecular bone tissues. In the revised manuscript, we included results showing that SELENOW-deficient mice [whole body $SELENOW^{-/-}$ and osteoclast-specific SELENOW-deficient mice ($LysM-Cre;SELENOW^{tm1c/tm1c}$)] exhibited decreased osteoclast size and eroded bone surface (Fig. 2c, f), but TG mice displayed increased osteoclast size and eroded bone surface (Fig. 3d), when compared with their wild type littermates.

Fig. R1. Apoptotic analysis of osteoclasts on the surface of trabecular bone in SELENOW-deficient ($SELENOW^{-/-}$) and SELENOW-overexpressing (TG) mice. Colour and fluorescent TUNEL staining with the TACS 2TdT DAB In Situ Apoptosis Detection Kit (Trevigen Inc., MD, USA; left panels) and In Situ Cell Death Detection Kit, AP (Cat.No.11 684 809 910, Roche, Germany; right panels) in the section of trabecular bone in $SELENOW^{-/-}$ (a) and TG mice (b). The images were visualized under light (left panel; the brown colour indicates apoptotic cell) and fluorescence microscope (right panel; the fluorescence dot indicates apoptotic cell). Scale bar, 100 μ m.

6) The authors state that it would be informative to perform additional studies in order to develop “anti-osteoporotic agents using Selenow”. Given the fact, that Selenow is primarily localized in the cytoplasm in a large number of different cell types or organs, this seems to be rather difficult. However, if the authors have specific ideas how their knowledge could translate into therapy, they should state this more precisely in the Discussion section.

Response: Thank you for your comment. We would like to clarify that we did not claim in our original manuscript that SELENOW could be used for the development of anti-osteoporotic agents in present. Instead, we described in the Discussion part as follows: Previous studies have reported that bone metastases in some

cancers, such as breast, prostate, and multiple myeloma, lead to the release of osteoclast-activating factors (e.g., β_2 -microglobulin, IL-1 β and TNF- α from myeloma cells, T cells, marrow stromal cells, and monocytes, thereby promoting osteoclastogenesis and osteoclastic bone resorption. Ria *et al.* also reported that bone marrow endothelial cells in active multiple myeloma patients with osteolytic lesions had notably induced *SELENOW* expression, which may influence disease progression. Since *SELENOW* may exert different effects depending on the disease type and severity, further studies are needed to investigate the specific role of *SELENOW* in various bone defects, including bone metastatic and osteoporosis-related cancers, for the development of anti-osteoporotic agents using *SELENOW*.

Reviewer #3 (Remarks to the Author):

This is an interesting manuscript detailing a potential role for RANKL induced downregulation of SELENOW promotes maintenance of proper osteoclast differentiation, while not affecting osteoblasts. There are some issues with controls, missed opportunities for in vivo insights, and technical approaches. Comments are below.

General issues:

1. For the gene chip arrays and RT-PCR, what was the up/down regulation compared to (csf only, no Rankl?)

Response: Thank you for your question. Gene expression levels of osteoclast precursors cultured with only M-CSF (30 ng/ml) were compared to those of osteoclast precursors cultured with both M-CSF (30 ng/ml) and RANKL (100 ng/ml), for gene chip arrays and RT-PCR. We added this statement in the section of Methods and the legend of Supplementary Fig. 1.

2. Western blot for selenow in figure 1a along with RNA levels, but why no western for the rest of the osteoclast specific markers?

Response: We conducted Western blotting for the osteoclast specific markers containing NFATc1, c-Fos, cathepsin K, and DC-STAMP. We added these results to Fig. 1a.

3. There are some peculiarities of the paper such as the use of RT-PCR followed by agarose gels instead of the current standard method of real-time PCR to quantify.

Response: In some figures of our manuscript, RT-PCR was performed to visualise and compare the expression level changes for several genes. We also analysed gene expression levels of *NFATc1*, *Acp5*, *OSCAR*, *RANKL* and *OPG* using real-time PCR (Supplementary Fig. 4e and Supplementary Fig. 10a, b). Thus, we do not have any specific bias for the use of RT-PCR.

4. The effects of SELENOW KO in vivo were surprisingly small for many of the figures (e.g. Fig 2d and e show about 30% less; parts of Fig 3 show small effects).

Response: Bone section analysis revealed approximately 38.1% fewer TRAP-positive osteoclasts on the surface of trabecular bone (NOc/BS) in SELENOW-deficient mice compared with wild type littermate controls (Fig. 2c; 6.151 ± 1.111 versus 3.806 ± 1.086). SELENOW-overexpressing transgenic mice had 36.9% more multinucleated TRAP-positive osteoclasts on the surface of trabecular bone than wild type littermate controls (Fig. 3d; 3.896 ± 0.441 versus 6.172 ± 1.118).

Besides functioning as a critical stimulator of positive osteoclastogenic factors including NFATc1, OSCAR, NF- κ B, Atp6v0d2, DC-STAMP, and c-Fos, RANKL is involved in a negative feedback auto-regulatory mechanism preventing hyperactive osteoclast activity and maintaining normal bone mass (Biochim Biophys Acta, 2014, 1840:3246-3256; Nature, 2002, 416:744-749; Endocrinology, 2005, 146:728-735). Two different types of a negative feedback mechanism by RANKL may occur through up-regulation of negative regulators or down-regulation of positive regulator during osteoclast differentiation. First, RANKL signalling itself was reported to induce the expression of two negative regulators for osteoclastogenesis, glutathione peroxidase-1 and interferon- β , which stimulates a negative feedback loop that contributes to bone mass homeostasis (Nature, 2002, 416:744-749; Endocrinology, 2005, 146:728-735). Second, in the present study, we revealed that SELENOW, a positive regulator of osteoclastogenesis, is gradually and significantly repressed during osteoclast differentiation, demonstrating that SELENOW plays a physiologically critical role in the negative feedback regulation of osteoclastogenesis and prevents hyperactive osteoclast activity and excessive bone resorption. Since there are various factors participating in negative feedback regulation of osteoclast differentiation *in vivo*, only the absence of SELENOW in osteoclasts may induce relatively small changes in bone parameters than expected. To note, SELENOW, a positive regulator of osteoclastogenesis, is required to be gradually down-regulated during osteoclast differentiation to maintain bone homeostasis in normal physiology.

In addition, to clearly demonstrate the effect of SELENOW on osteoclasts and bone metabolism, we

generated osteoclast-specific SELENOW-deficient mice (LysM-Cre;SELENOW^{tm1c/tm1c}). Microcomputed tomography (μ CT) analysis of trabecular bone in the proximal tibia demonstrates that osteoclast-specific LysM-Cre;SELENOW^{tm1c/tm1c} mice showed an increase in bone mass resulting from elevated trabecular bone mineral density, bone volume, trabecular number and trabecular thickness, and decreased trabecular bone separation when compared with wild-type littermate mice (SELENOW^{tm1c/tm1c}), as presented in Fig. 2d-g. Consistently, histomorphometric analysis of LysM-Cre;SELENOW^{tm1c/tm1c} mice exhibited a decrease in osteoclast number, osteoclast size, and eroded bone surface on the surface of trabecular bone when compared with littermate control mice (SELENOW^{tm1c/tm1c}). Taken together, our new results demonstrate that mice with SELENOW deficiency in osteoclasts showed an increase in bone mass, which is consistent with the osteoprotective phenotype of whole-body SELENOW knockout mice (SELENOW^{-/-} mice).

5. I don't really see how this study shows molecular mechanism, other than binding to certain proteins and somehow SELENOW reduces cytosolic redox status. Is the role of SELEONW to prevent oxidation of specific binding partners? The way experiments are conducted all of the effects could be either direct or indirect.

Response: Selenoproteins contain a selenocysteine (SeCys) amino acid residue translated from in-frame UGA codon that is normally recognized as a stop codon (Nature, 2000, 407:463-465). The SeCys insertion sequence (SECIS) element in the 3' untranslated region (UTR) of eukaryotic selenoprotein-encoding mRNAs is an RNA element approximately 60 nucleotides in length and has a conserved stem-loop structure, which is essential for the decoding of UGA codon as a SeCys. Because the secondary structure of the SECIS element in SELENOW mRNA is required for the successful translation of UGA as SeCys and since Selenoprotein W is the smallest mammalian selenoprotein (89 amino acids), it is very difficult to construct a series of sequential SELENOW deletion mutants lacking NFATc1-interacting domain and NF- κ B-interacting domain as well as an antioxidant domain. Instead, we thought that a SeCys at codon 13 plays an important role in the interaction of SELENOW with NFATc1 and NF- κ B. To investigate the interaction of SELENOW with NFATc1 and NF- κ B, we used cells with SELENOW mutant in which a SeCys is substituted with a cysteine (SeCys13C) or a serine (SeCys13S). Both mutants, SELENOW (SeCys13C) and SELENOW (SeCys13S), were unable to interact with either NFATc1 or NF- κ B (Fig. 4d), indicating that a SeCys at codon 13 plays a critical role in the interaction between SELENOW and its targets with NFATc1 and NF- κ B. In addition, we previously showed that a SeCys at codon 13 is crucial for the antioxidant activity of SELENOW (FEBS Letters, 2002, 517:225-228). Together, a SeCys at codon 13 of SELENOW is thought to be critical for the interaction of SELENOW with NFATc1 and NF- κ B as well as for its antioxidant activities.

Based on the result which indicates that a SeCys-13 residue of SELENOW plays a part in the binding of SELENOW to NFATc1 or NF- κ B, as well as new additional experimental data at the revision stage and previously reported, we described a possible mechanistic role of SELENOW in osteoclast differentiation in the revised Discussion as follows:

“A specific interaction between SELENOW and 14-3-3 protein is known to involve the conserved redox motif CysXXSeCys present in a SELENOW exposed loop, and an exposed Cys residue in the C-terminal domain of 14-3-3 protein (Biochemistry, 2007, 46:6871-6882). The binding of SELENOW to 14-3-3 protein was significantly enhanced under oxidative stress conditions (Biochim Biophys Acta, 2016, 1863:10-18), indicating that the intracellular redox status regulates SELENOW-14-3-3 protein interactions, and modulates 14-3-3 protein functional activity. Here, we observed that the SELENOW SeCys-13 residue is critical for the interaction with its targets, NFATc1 and NF- κ B, and that SELENOW leads to an increase in cell reduction capacity. Also, we previously highlighted that this residue, SeCys-13, is crucial for SELENOW antioxidant activity (FEBS Letters, 2002, 517:225-228). Altogether, these findings suggest that SELENOW has the potential to enhance osteoclast differentiation via direct interaction with NFATc1 and NF- κ B and their subsequent activation, but also indirectly, as a reducing agent.”

Additionally, we have indicated “negative feedback mechanism of osteoclastogenic SELENOW in osteoclast differentiation” in the first section of the Discussion. We hope the reviewer will accept the rationale of our work upon assessing our entire manuscript as a whole.

6. For the in vivo portion there were issues, and properly controlled experiments would allow better

interpretation. For instance, in humans there are large discrepancies in rates of osteoporosis in males and females, I would have liked to have seen both sexes separately used (although they did specify they used male mice). Furthermore, bone density can be highly positively correlated with body mass, I would like to have seen controls for body weight. And also they should have compared litter mates as is required for top tier journals like Nature Communications.

Response: As the reviewer recommended, we analysed the role of SELENOW in physiological bone remodelling in SELENOW-deficient and SELENOW-overexpressing female mice, which showed that the results obtained from female mice were almost identical to those from male mice. We added this data to Supplementary Fig. 6 and 9.

Bone mineral density has been reported to be closely associated with body mass index (JCEM, 2014, 99:30-38; AO, 2014, 9:175). As presented in Supplementary Fig. 5a and 8a, body weight of ten-week-old SELENOW-deficient male mice was higher than that of wild type littermates (27.77 ± 1.06 g versus 24.65 ± 1.74 g, $P < 0.01$) and SELENOW-overexpressing mice exhibited a reduced body weight compared with their wild type littermates (28.01 ± 1.63 g versus 30.91 ± 2.23 g, $P < 0.01$). Our results indicate that bone mineral density is highly positively correlated with body weight in SELENOW-deficient mice and SELENOW-overexpressing mice. This was described in the Result section of Supplementary Fig. 5a and 8a.

In addition, in this study, the littermates from SELENOW-deficient mice, SELENOW-overexpressing mice, and their wild type mice were analyzed and we added this statement in Methods section.

7. Fig. 5 g, h, i all show that when selenow is overexpressed, there is an observed increase in reduction capacity in osteoclasts and increased survival (well they don't do an apoptosis assay but instead just count number of cells compared to control). The total thiol experiments in Fig. 5 involve unusually high levels of SELENOW and likely to not reflect physiologically relevant roles for the endogenous protein. Overexpression of any selenoprotein is known to do this, and the investigators need to come up with a better way to determine molecular mechanisms.

Response: We described a possible mechanistic role of SELENOW in osteoclast differentiation in the response to question #5 and would appreciate it if the reviewer would reconsider this point. Moreover, we tried to analyse the change of osteoclastic characteristics in the absence or presence of SELENOW *in vivo*. **First**, we performed the assay of osteoclastic cell apoptosis on the surface of trabecular bone from SELENOW-deficient ($SELENOW^{-/-}$) and SELENOW-overexpressing (TG) mice. For the detection of apoptotic osteoclast cells, after the tibial trabecular bone from $SELENOW^{-/-}$ and TG mice was decalcified for 2 weeks, 5- μ m-thick sagittal sections were prepared on a microtome and we performed *in vivo* terminal deoxynucleotidyl transferase-mediated deoxyuridine triphosphate nick end labeling (TUNEL) assay using the TACS 2TdT DAB In Situ Apoptosis Detection Kit (Trevigen Inc., MD, USA) and In Situ Cell Death Detection Kit, AP (Cat.No.11 684 809 910, Roche, Germany). The colour and fluorescence sections stained with TUNEL were visualized under respective light and fluorescence microscope (Fig. R1). Also, the sequential sections were stained with TRAP to detect osteoclasts (data not shown). Unfortunately, we could not discriminate the apoptotic osteoclasts from other numerous non-osteoclastic cells in overlapped images between TUNEL-positive apoptotic cells and TRAP-positive osteoclasts. Some reasons for detecting apoptotic osteoclasts in bone sections include: i) osteoclasts are specialized multinucleated giant cells derived from monocyte fusion and have about 2 to 100 nuclei per cell (Endoc Rev, 1996, 17:308-332), and it is difficult to distinguish the apoptotic images of osteoclasts with dispersed multi-nuclei from those of non-osteoclastic cells with mono-nucleus. ii) As previously reported (Endoc Rev, 2000, 21:115-137), the terminally differentiated mature osteoclasts that resorb the bone have a short life span of approximately 2 weeks, whereas the life span of osteoblasts is approximately 3 months. A short retention time of osteoclasts *in vivo* will make it difficult to observe the apoptotic osteoclasts. iii) Decalcification process for a period of 2 weeks to produce bone sections may cause destruction of apoptotic factors and interfere with the detection of apoptotic osteoclast cells. **Second**, instead of apoptotic data of osteoclasts, we analysed osteoclast size and eroded bone surface, which reflect the existence of bone-resorbing osteoclasts and osteoclastic function, in trabecular bone tissues. In the revised manuscript, we included results showing that SELENOW-deficient mice [whole body $SELENOW^{-/-}$ and osteoclast-specific SELENOW-deficient mice ($LysM-Cre;SELENOW^{tm1c/tm1c}$)] exhibited decreased osteoclast size and eroded bone surface (Fig. 2c, f), but TG mice displayed increased osteoclast size and eroded bone surface (Fig. 3d), when compared with their wild type littermates. We, to the best of our ability, provided an evidentiary link between SELENOW and osteoclast differentiation. We hope that the reviewer will accept the rationale of our work.

Fig. R1. Apoptotic analysis of osteoclasts on the surface of trabecular bone in SELENOW-deficient ($SELENOW^{-/-}$) and SELENOW-overexpressing (TG) mice. Colour and fluorescent TUNEL staining with the TACS 2TdT DAB In Situ Apoptosis Detection Kit (Trevigen Inc., MD, USA; left panels) and In Situ Cell Death Detection Kit, AP (Cat.No.11 684 809 910, Roche, Germany; right panels) in the section of trabecular bone in $SELENOW^{-/-}$ (a) and TG mice (b). The images were visualized under light (left panel; the brown colour indicates apoptotic cell) and fluorescence microscope (right panel; the fluorescence dot indicates apoptotic cell). Scale bar, 100 μ m.

8. There seems to be missed opportunities on studying the mice involved in this manuscript for bone disease. The tissues were mainly studied ex vivo, but are there stress tests or other in vivo tests that could provide interesting insights into the outcome of loss-of-function and gain-of-function of SELENOW in mice?

Response: Thank you for your comment. We analysed more detailed phenotypes of SELENOW-deficient and SELENOW-overexpressing mice. No significant differences were observed in whole-body X-ray images of the skeleton, tooth eruption, as well as calcium and phosphate serum levels between the SELENOW-deficient ($SELENOW^{-/-}$) mice and wild type littermates (Supplementary Fig. 5). SELENOW-overexpressing mice had also no significant changes in whole-body X-ray images, tooth eruption, and serum levels of calcium and phosphate when compared with their wild type littermates (Supplementary Fig. 8).

Body weight of $SELENOW^{-/-}$ mice was higher than that of wild type littermates (27.77 ± 1.06 g versus 24.65 ± 1.74 g, $P < 0.01$; Supplementary Fig. 5a), and SELENOW-overexpressing mice exhibited a reduced body weight compared with their wild type littermates (28.01 ± 1.63 g versus 30.91 ± 2.23 g, $P < 0.01$; Supplementary Fig. 8a).

To investigate neuromotor coordination and balance, mice were analysed by rotarod experiments. In the accelerating rotarod, no significant differences in latency to fall from the test apparatus were observed between SELENOW-deficient mice and wild type littermates as well as between SELENOW-overexpressing mice and wild type littermates (Supplementary Fig. 5d and 8d). We also performed the open field test to study the overall locomotor activities of SELENOW-deficient and SELENOW-overexpressing mice. The total travel distances were not significantly different between SELENOW-deficient mice and wild type littermates as well as between SELENOW-overexpressing mice and wild type littermates (Supplementary Fig. 5d and 8d).

Specific comments

1. Line 47, 56, 61, 147: replace semi-colon with a period.

Response: As suggested, we replaced semi-colon with a period.

2. Line 88: Need one or two more sentences describing the GeneChip arrays. What type of cells? How were they stimulated for how long? What was compared (e.g. unstimulated s stimulated)?

Response: The gene expression levels of osteoclast precursors cultured with only M-CSF (30 ng/ml) were compared to those of osteoclast precursors treated with both M-CSF (30 ng/ml) and RANKL (100 ng/ml), for the GeneChip arrays and RT-PCR. M-CSF (30 ng/ml) and RANKL (100 ng/ml) were stimulated for 24 h, 48 h, or 72 h. We included this in the legend of Supplementary Fig. 1a and b.

3. The data in Fig. 1b are not conclusive and should be performed using real-time PCR. Also, the lack of an effect of IFN γ on SELENOW levels may be due to mistiming of adding the IFN γ or other aspects of the experimental approach. Were different pre-incubation periods with IFN γ attempted before the RANKL was added. These types of negative data are not conclusive as presented without controls that rule out procedural artifacts.

Response: Interferon- γ has been reported to be an inducer of TRAF6 proteasomal degradation (Nature, 2000, 408:600-605). In our study, RANKL treatment for 24 h or 48 h resulted in a significant reduction of SELENOW expression in the absence of interferon- γ , showing the RANKL/RANK/TRAF6 axis-dependent downregulation of SELENOW (Fig. 1b). However, pretreatment of osteoclast precursors with interferon- γ (150 U/ml) 30 min prior to RANKL stimulation caused an increased level of SELENOW and failed to induce RANKL-mediated SELENOW downregulation, which suggests that interferon- γ promoted the proteasomal degradation of TRAF6 and thus successfully inhibited RANKL/RANK/TRAF6 signalling. Treatment of osteoclast precursors with interferon- γ clearly upregulated SELENOW levels when compared with untreated control cells (Fig. 1b). Therefore, we consider our data to be conclusive and are not negative.

4. Line 137; data not shown should be submitted as data in this paper. Characterization of these mice is crucial for the following experiments and the reader needs to be convinced the tissues are affected by the genetic manipulations employed. Need to show a tissue blot with various tissues analyzed for SELENOW.

Response: As the reviewer commented, we included results demonstrating the SELENOW deficiency in any tissue of SELENOW^{-/-} mice or during RANKL-induced osteoclast differentiation of SELENOW^{-/-} mouse-derived osteoclast precursors in Supplementary Fig. 4a and c of the revised manuscript.

5. Fig. 2b, real-time PCR for mRNA relative to which housekeeping mRNA?

Response: The mRNA level was normalized to that of glyceraldehyde 3-phosphate dehydrogenase and this statement was already described in the Methods section of our original manuscript.

6. LINE 177 change signalling to signaling.

Response: As suggested, we changed signalling to signaling.

7. Fig 4a shows nuclear vs. cytosolic NF κ B, but need to do EMSA to show it is functional. The reporter assays in 4b support this idea, but EMSA is the gold standard for NF κ B translocation and binding to DNA.

Response: As the reviewer commented, the electrophoretic mobility shift assay (EMSA), also called a gel shift assay, is a very popular technique to detect protein-DNA interactions and is the golden standard assay for the presence of NF- κ B that is capable of binding DNA. Chromatin immunoprecipitation (ChIP) assay, is also widely used to assess NF- κ B binding to the promoters and enhancers of specific genes. We have demonstrated the nuclear translocation and binding to target DNA of osteoclastogenic transcription factors NF- κ B and NFATc1 using Western blotting analysis in cytosolic and nuclear fractionation, luciferase reporter assay, and ChIP assay.

8. Line 188 artefact change to artifact.

Response: Thank you for your suggestion. We changed artefact to artifact.

9. Line 251 “redox status was reduced...” is not clear. Do you mean shifted toward an oxidized state?

Response: Thank you for your comment. We revised “redox status was reduced after mature osteoclast formation” to “redox status was shifted toward oxidised state after mature osteoclast formation”.

10. Line 254 “SEELENOW” change to SELENOW.

Response: Thank you for your comment and we apologize for the typo, which we have corrected.

Reviewers' Comments:

Reviewer #1:

Remarks to the Author:

In the revised manuscript, the authors generated SELENOW conditional knockout mice and showed that SELENOW function is osteoclast-intrinsic. This data improved the revised manuscript significantly, but the authors first need to show the cell-type specific deletion of SELENOW in the osteoclast lineage.

The authors failed to provide sufficient evidence showing that the gradual reduction of SELENOW is essential for osteoclast differentiation. The analysis of conditional knockout mice is incomplete and it is unclear whether they developed authentic osteopetrosis. The molecular mechanism underlying the SELENOW function is not well shown.

Importantly, the authors failed to prove the importance of "gradual repression" of SELENOW in bone homeostasis. As the title of this paper indicated, the main concept of this study is that the down-regulation of SELENOW is required for the proper osteoclastogenesis, but this is not supported by the presented data.

How SELENOW regulates osteoclast differentiation pathways in a down-regulation dependent manner should be shown. Otherwise, the title does not match the content. Precise molecular mechanisms are needed to meet the high criteria for publication in Nature Communications.

Major concerns

1. Although the authors showed that SELENOW directly interacts with NFATc1 and NF- κ B, it is still unclear how the direct interaction of SELENOW with these molecules contributes to osteoclastogenesis.
2. The authors stated that SELENOW-deficient and SELENOW-conditional knockout mice developed osteopetrosis. To prove this, the authors should show whether mice had tooth eruption and contained cartilage remnants. To this reviewer, the mice seem to have a high bone mass phenotype but not "authentic" osteopetrosis. The authors should provide the histological data (osteoclasts and bone volume, in particular) and complete parameters of bone morphometric analysis of SELENOW-conditional knockout mice.
3. The authors claimed that the "gradual repression" of SELENOW is essential for bone homeostasis. This hypothesis is based on the findings that the expression levels of SELENOW is down-regulated during osteoclastogenesis and the SELENOW-transgenic mice decreased bone volume due to an enhanced osteoclastogenesis. Since the SELENOW-transgenic mice constitutively express aberrant levels of SELENOW in TRAP-positive cells, this system is too artificial and cannot prove the importance of the "gradual repression" of SELENOW in physiological bone remodeling. The authors should utilize more accurate methods to prove their concept or, alternatively, change the title to reflect their findings more properly.
4. There are some inconsistencies in the presented data. In figures 2b and 2c, the bone volume seems not decrease in SELENOW knockout mice. In figures 2a and 2d, micro CT picture showed that the high bone mass phenotype is severer in SELENOW conditional KO mice than SELENOW KO mice. More representative images are warranted to allow correct interpretation of the data.

Reviewer #2:

Remarks to the Author:

The authors have adequately addressed all of my comments and further improved their manuscript.

Reviewer #3:

Remarks to the Author:

The authors have done a good job of addressing issues raised by all reviewers. The only concern I still have is the use of LysM-cre to KO SELENOW in osteoclasts, when other mice have more traditionally been used including CD11b-cre. However, the new data support KO of SELENOW in osteoclasts using the LysM-cre, so this is a minor concern. Overall, the issues have been addressed.

Response to referees
(NCOMMS-18-01627B)

Reviewer #1 (Remarks to the Author):

In the revised manuscript, the authors generated SELENOW conditional knockout mice and showed that SELENOW function is osteoclast-intrinsic. This data improved the revised manuscript significantly, but the authors first need to show the cell-type specific deletion of SELENOW in the osteoclast lineage.

The authors failed to provide sufficient evidence showing that the gradual reduction of SELENOW is essential for osteoclast differentiation. The analysis of conditional knockout mice is incomplete and it is unclear whether they developed authentic osteopetrosis. The molecular mechanism underlying the SELENOW function is not well shown.

Importantly, the authors failed to prove the importance of “gradual repression” of SELENOW in bone homeostasis. As the title of this paper indicated, the main concept of this study is that the down-regulation of SELENOW is required for the proper osteoclastogenesis, but this is not supported by the presented data.

How SELENOW regulates osteoclast differentiation pathways in a down-regulation dependent manner should be shown. Otherwise, the title does not match the content. Precise molecular mechanisms are needed to meet the high criteria for publication in Nature Communications.

Answer: We appreciate the reviewer’s considerable comments. The reviewer firstly indicated that our data should show the cell type-specific deletion of selenoprotein W in the osteoclast lineage. As shown in Figure S4b, our previous result showed that osteoclast-specific SELENOW-deficient mice (LysM-Cre; SELENOW^{tm1c/tm1c}) express selenoprotein W in various tissues tested, but not in osteoclast precursors differentiated from bone marrow-derived monocytes. Thus, this issue was concretely described in the second subsection of the “Results” section and the legend of Supplementary Fig. 4b.

Actually, a unique gene in osteoclasts, multinucleated bone resorbing cells that originate from hematopoietic stem cells and develop through the fusion of myeloid-lineage mononuclear cells, was specifically deleted using various Cre models such as Mx1-Cre, LysM-cre, CD11b-Cre, c-Fms-Cre, RANK-Cre, TRAP-Cre, and Ctsk-Cre (*J Clin Invest* 2008, 18:3775-3789; *Curr Osteoporos Rep* 2018, 16:466-477). Mx1-Cre and LysM-cre are targeted in myeloid-lineage cells, CD11b-Cre, c-Fms-Cre, and RANK-Cre in macrophages, and TRAP-Cre and Ctsk-Cre in mature osteoclasts. However, each model has its own limitation (*Curr Osteoporos Rep* 2018, 16:466-477). Considering that selenoprotein W was highly expressed in the earliest stage of osteoclasts, we selected the LysM-Cre/loxP system to confirm osteoclast-intrinsic function of selenoprotein W in bone remodeling in the hematopoietic myeloid-osteoclast lineage. In previous reports, the LysM-Cre/loxP system was used for osteoclast-targeted gene deletion (*J Bone Mineral Research* 2015, 30:1138-1149; *Bone Research* 2020, 8:11; *Nature Medicine* 2016, 22:1203-1205). We would like the reviewer to consider the limitations of each mouse model.

Furthermore, we have outlined the answers to other major concerns as follows.

Major concerns

1. Although the authors showed that SELENOW directly interacts with NFATc1 and NF- κ B, it is still unclear how the direct interaction of SELENOW with these molecules contributes to osteoclastogenesis.

Answer: We appreciate your valuable comment, and agree with your suggestion. In this study, we showed that SELENOW interacts with NFATc1 and NF- κ B, and a SeCys at codon 13 of SELENOW residue is critical for its interaction with either NF- κ B or NFATc1. SELENOW promoted the nuclear translocation of cytosolic NF- κ B and NFATc1. In addition, we have demonstrated the stimulatory effect of SELENOW on NF- κ B-dependent and NFATc1-dependent transcriptional activity using luciferase reporter assay and ChIP assay. It has been well established that translocation of NF- κ B to the nucleus induces expression of c-Fos and NFATc1 (*Nature* 2003, 423:337–342). In addition, translocation of NFATc1 to the nucleus regulates osteoclast specific genes such as TRAP, calcitonin receptor, cathepsin K, β 3 integrin, OSCAR, DC-STAMP, matrix metalloproteinase 9 (MMP9), ATPase H⁺ transporting V0 subunit d2 (Atp6v0d2), and c-Src (*J Exp Med* 2004, 200:941-946; *Immunol Rev* 2009, 231:241-256; *FEBS Lett* 2009, 583:2435-2440; *Int J Biochem Cell Biol* 2010, 42:576-579; *J Bone Metab* 2014, 21:233-241). Consistently, we showed that during osteoclast differentiation, c-Fos, TRAP, and cathepsin K were upregulated in SELENOW-overexpressing osteoclasts when compared with that in the wild type osteoclasts (Supplementary Fig. 7). Collectively, our manuscript at the first stage of revision showed that the interaction of SELENOW with NFATc1 and NF- κ B results in the nuclear translocation and activation of NFATc1 and NF- κ B, and then induction of osteoclast-specific genes, contributing to osteoclastogenesis.

Now, at the second stage of revision, we performed more experiments to investigate how the interaction of SELENOW with NFATc1 and NF- κ B contributes to osteoclastogenesis. To further validate SELENOW-induced activation of NF- κ B and NFATc1, we studied the nuclear translocation of cytosolic NF- κ B and NFATc1 in SELENOW-deficient and SELENOW-overexpressing preosteoclasts in response to RANKL treatment. RANKL-induced nuclear translocation of NF- κ B and NFATc1 increased in SELENOW-overexpressing preosteoclasts, but decreased in SELENOW-deficient preosteoclasts compared with that in the wild type preosteoclasts, confirming that SELENOW specifically interacts with and induces nuclear translocation of NF- κ B and NFATc1 via 14-3-3 γ . This new data is now included in Fig. 4f and Supplementary Fig. 13a.

The 14-3-3 proteins have been reported to play a critical role in the regulation of several biological processes such as cell cycle, cell growth, differentiation, survival, apoptosis, metabolism, protein trafficking, stress responses, energy and nutrient homeostasis, and malignant transformation (*J Cell Sci* 2004, 117:1875-1884; *Trends Cell Biol* 2009, 19:16-23). Binding of 14-3-3 proteins to their targets has been shown to change the localization, stability, phosphorylation state, and activity of a target protein (*J Cell Sci* 2004, 117:1875-1884). In particular, 14-3-3 proteins have been reported to participate in various cellular pathways through altering the subcellular localization of their binding partners (*Cell Signal* 2000, 12:703-709). 14-3-3 proteins have been implicated in cytoplasm-nucleus shuttling, cytoplasm-mitochondrion shuttling, endoplasmic reticulum-plasma membrane trafficking, and nuclear retention of target proteins (*Cell Signal* 2000, 12:703-709). For example, 14-3-3 was shown to enhance nuclear localization of thioredoxin-like protein 2, checkpoint kinase 1, telomerase reverse transcriptase, and T-cell protein tyrosine phosphatase (*EMBO J* 2000, 19:2652-2661; *J Biol Chem* 1998, 273:25356-25363; *Oncotarget* 2017, 8:90674-90692; *J Biol Chem* 2003, 278:25207-25217).

It has been reported that SELENOW interacts specifically with 14-3-3 protein, which has been reported to interact with NFAT proteins and NF- κ B (*J Biol Chem* 2007, 282:37036-37044; *J Proteome Res* 2011, 10:968-976, *Mol Cell Biol* 2000, 20:702-712; *J Cell Sci* 2006, 119:3695-3704; *PLOS One* 2012, 7:e38347). We have also showed that 14-3-3 γ formed a complex with SELENOW and was translocated to the nucleus (Fig. 4a, c, f and Supplementary Fig. 12). In addition, as shown in Fig. 4c and Supplementary Fig. 12, SELENOW formed a complex with NF- κ B, NFATc1, and 14-3-3 γ . Moreover, nuclear translocation of 14-3-3 γ and SELENOW was increased in SELENOW-overexpressing preosteoclasts compared with that in the wild type preosteoclasts (Fig. 4f; left panel). Based on these results, we hypothesized that 14-3-3 γ plays an important role in the nuclear translocation of NF- κ B and NFATc1 induced by SELENOW.

To investigate the functional role of 14-3-3 γ in SELENOW-mediated nuclear translocation of NF- κ B and NFATc1, we performed 14-3-3 γ knockdown through lentiviral-delivered shRNAs. In wild type preosteoclasts, 14-3-3 γ -deficient cells showed decreased translocation of NF- κ B and NFATc1 to the nucleus after RANKL treatment compared to that in the control cells (Supplementary Fig. 13d). Consistently, in SELENOW-overexpressing preosteoclasts, depletion of 14-3-3 γ resulted in a significant decrease in the translocation of SELENOW as well as NF- κ B and NFATc1 to the nucleus in response to RANKL treatment, when compared to that in the control cells (Fig. 4f; right panel). Furthermore, we studied the effects of 14-3-3 γ depletion on the differentiation of SELENOW-overexpressing osteoclast precursors into multinucleated osteoclasts. 14-3-3 γ depletion resulted in a decrease in the formation of TRAP-positive multinucleated cells (Fig. 4g). Together, our results clearly indicate that 14-3-3 γ mediates nuclear translocation of NF- κ B and NFATc1 induced by SELENOW, contributing to osteoclastogenesis. Further study on the interaction of SELENOW with NF- κ B and NFATc1 through 14-3-3 protein will extend our understanding of the regulation of osteoclastogenesis by SELENOW. We have described these findings in the Results and Discussion sections of the manuscript.

We hope that the reviewer will accept the rationale for the functional study of SELENOW in osteoclast differentiation and bone remodeling.

2. The authors stated that SELENOW-deficient and SELENOW-conditional knockout mice developed osteopetrosis. To prove this, the authors should show whether mice had tooth eruption and contained cartilage remnants. To this reviewer, the mice seem to have a high bone mass phenotype but not “authentic” osteopetrosis. The authors should provide the histological data (osteoclasts???? and bone volume, in particular) and complete parameters of bone morphometric analysis of SELENOW-conditional knockout mice.

Answer: As the reviewer suggested, we have now clarified that SELENOW-deficient and SELENOW-conditional knockout mice showed a high bone mass phenotype, but not osteopetrosis. This has been described in the text throughout the revised manuscript. In addition, we have included the histological data for osteoclasts and bone volume, and the parameters of bone morphometric analysis in SELENOW-conditional knockout mice (Fig. 2e, f).

3. The authors claimed that the “gradual repression” of SELENOW is essential for bone homeostasis. This hypothesis is based on the findings that the expression levels of SELENOW is down-regulated during osteoclastogenesis and the SELENOW-transgenic mice decreased bone volume due to an enhanced

osteoclastogenesis. Since the SLENOW-transgenic mice constitutively express aberrant levels of SELENOW in TRAP-positive cells, this system is too artificial and cannot prove the importance of the “gradual repression” of SELENOW in physiological bone remodeling. The authors should utilize more accurate methods to prove their concept or, alternatively, change the title to reflect their findings more properly.

Answer: As the reviewer pointed out, we feel that our results do not offer sufficient evidence regarding the critical role of gradual repression of SELENOW in regulation of osteoclast development and bone remodeling. Thus, we have deleted the word “gradual repression” that explains the functional role of SELENOW in osteoclastogenesis and bone remodeling, and would like to change the previous title to “Selenoprotein W ensures physiological bone remodeling by preventing hyperactivity of osteoclasts”. We hope that the new title is appropriate.

4. There are some inconsistencies in the presented data. In figures 2b and 2c, the bone volume seems not decrease in SELENOW knockout mice. In figures 2a and 2d, micro CT picture showed that the high bone mass phenotype is severer in SELENOW conditional KO mice than SELENOW KO mice. More representative images are warranted to allow correct interpretation of the data.

Answer: In Fig. 2, we showed that SELENOW^{-/-} mice had increased bone mass resulting from increases in trabecular bone volume, number, thickness, and mineral density. Therefore, in Fig. 2b and 2c, the bone volume increased in SELENOW knockout mice. In addition, we have replaced μ CT images with a more representative one in Fig. 2a and 2d.

Reviewer #2 (Remarks to the Author):

The authors have adequately addressed all of my comments and further improved their manuscript.

Answer: We thank the reviewer for the positive response.

Reviewer #3 (Remarks to the Author):

The authors have done a good job of addressing issues raised by all reviewers. The only concern I still have is the use of LysM-cre to KO SELENOW in osteoclasts, when other mice have more traditionally been used including CD11b-cre. However, the new data support KO of SELENOW in osteoclasts using the LysM-cre, so this is a minor concern. Overall, the issues have been addressed.

Answer: Thank you again for your critical comment. The unique gene targeted in osteoclasts, multinucleated bone resorbing cells derived from mononuclear hematopoietic myeloid-lineage cells, was specifically deleted

using various Cre models such as Mx1-Cre, LysM-cre, CD11b-Cre, c-Fms-Cre, RANK-Cre, TRAP-Cre, and Ctsk-Cre (*J Clin Invest* 2008, 18:3775-3789; *Curr Osteoporos Rep* 2018, 16:466-477). Mx1-Cre and LysM-cre are targeted in myeloid-lineage cells, CD11b-Cre, c-Fms-Cre, and RANK-Cre in macrophages, and TRAP-Cre and Ctsk-Cre in mature osteoclasts. However, each model has its own limitation (*Curr Osteoporos Rep* 2018, 16:466-477). Considering that selenoprotein W was highly expressed in the earliest stage of osteoclasts, we selected the LysM-Cre/loxP system to confirm osteoclast-intrinsic function of selenoprotein W in bone remodeling in the hematopoietic myeloid-osteoclast lineage. In previous reports, the LysM-Cre/loxP system was used for osteoclast-targeted gene deletion (*J Bone Mineral Research* 2015, 30:1138-1149; *Bone Research* 2020, 8:11; *Nature Medicine* 2016, 22:1203-1205). We would like the reviewer to consider the limitations of each mouse model.

Reviewers' Comments:

Reviewer #1:

Remarks to the Author:

The authors appropriately addressed all of my comments and improved the manuscript. No further revisions are needed.